# Modeling Causal Mechanisms with Diffusion Models for Interventional and Counterfactual Queries

**Patrick Chao**[*]                                                                          *pchao@wharton.upenn.edu*
*University of Pennsylvania*

**Patrick Blöbaum**                                                                          *bloebp@amazon.com*
*Amazon*

**Sapan Patel**                                                                          *sapanp@amazon.com*
*Amazon*

**Shiva Prasad Kasiviswanathan**                                                                          *kasivisw@gmail.com*
*Amazon*

**Reviewed on OpenReview:** *https://openreview.net/forum?id=EDHQDsqiSe&referrer*

## Abstract

We consider the problem of answering observational, interventional, and counterfactual queries in a causally sufficient setting where only observational data and the causal graph are available. Utilizing the recent developments in diffusion models, we introduce diffusion-based causal models (DCM) to learn causal mechanisms, that generate unique latent encodings. These encodings enable us to directly sample under interventions and perform abduction for counterfactuals. Diffusion models are a natural fit here, since they can encode each node to a latent representation that acts as a proxy for exogenous noise. Our empirical evaluations demonstrate significant improvements over existing state-of-the-art methods for answering causal queries. Furthermore, we provide theoretical results that offer a methodology for analyzing counterfactual estimation in general encoder-decoder models, which could be useful in settings beyond our proposed approach.

## 1 Introduction

Understanding the causal relationships in complex problems is crucial for making analyses, conclusions, and generalized predictions. To achieve this, we require causal models and queries. Structural Causal Models (SCMs) are generative models describing the causal relationships between variables, allowing for observational, interventional, and counterfactual queries (Pearl, 2009a). An SCM specifies how a set of endogenous (observed) random variables is generated from a set of exogenous (unobserved) random variables with prior distribution via a set of structural equations.

Given a causal DAG, we focus on approximating the individual SCMs with diffusion models. As an application, we present a flexible framework for answering all the three types of causal (observational, interventional, and counterfactual) queries. For counterfactuals, we work in Pearl's SCM framework (Pearl, 2009a), and seek to quantify unit-level statements of the form: Given that observed a factual sample $(x_1^{\mathrm{F}}, \ldots, x_K^{\mathrm{F}})$ for a set of $K$ variables $(X_1, \ldots, X_k)$, what would have been the outcome for these $K$ variables be, if the value of some set $X_{\mathcal{I}}$ (with $I \subseteq [K]$) had been set to some $\gamma \in \mathbb{R}^{|\mathcal{I}|}$? Throughout this paper we assume *causal sufficiency*, i.e., absence of hidden confounders. Note that causal sufficiency is a necessary assumption for answering causal queries from observational data alone.

In the SCM framework, causal queries can be answered by learning a proxy for the unobserved exogenous noise and the structural equations. This suggests that (conditional) *generative models* that encode to a latent space could be an attractive choice for the modeling SCMs, as the latent serves as the proxy for the exogenous noise. In these models,

---

[*]Work done during internship at Amazon.

the encoding process extracts the latent from an observation, and the decoding process generates the sample from the latent, approximating the structural equations.

**Our Contributions.** In this work, we propose and analyze the effectiveness of using a diffusion model for modeling SCMs. Diffusion models (Sohl-Dickstein et al., 2015; Ho et al., 2020; Song et al., 2021) have gained popularity recently due to their high expressivity and exceptional performance in generative tasks (Saharia et al., 2022; Ramesh et al., 2022; Kong et al., 2021). The primary contribution of our work is in determining how to apply diffusion models to the causal setting. Rather than focusing a single causal inference setting, our aim through this modeling is to provide a single flexible framework that works for answering a wide range of causal queries. Now applying diffusion models to the causal setting is non-trivial because diffusion models are typically used to learn a stochastic process mapping between a standard Gaussian and a data distribution. Our main idea is to model each node in the causal graph as a diffusion model and cascade generated samples in topological order to answer causal queries. For each node, the corresponding diffusion model takes as input the node and parent values to encode and decode a latent representation. To implement the diffusion model, we utilize the recently proposed *Denoising Diffusion Implicit Models* (DDIMs) (Song et al., 2021), which may be interpreted as a deterministic autoencoder model without any dimensionality reduction while encoding. We leverage the deterministic forward and backward diffusion processes of DDIMs to use diffusion models as an encoder-decoder model. We refer to the resulting model as *diffusion-based causal model* (DCM) and show that this model mimics the necessary properties of an SCM. Our key contributions include:

**(1)** [Section 3] We propose diffusion-based causal model (DCM), a new model class for modeling structural causal models, that provides a flexible and practical framework for approximating both interventions (do-operator) and counterfactuals (abduction-action-prediction steps). We present a procedure for training a DCM given just the causal graph and observational data, and show that the resulting trained model enables sampling from the observational and interventional distributions, and facilitates answering counterfactual queries.

**(2)** [Section 4] Our theoretical analysis examines the accuracy of counterfactual estimates generated by the DCM, and we demonstrate that they can be bounded given some reasonable assumptions. Importantly, our analysis is not limited to diffusion models, but also applies to other encoder-decoder settings. To the best of our knowledge, these are the first error bounds that explain the observed performance improvements in using encoder-decoder models, like diffusion models, to address counterfactual queries. Another feature of this result, is that it also extends, under an additional assumption, to the more challenging multivariate case.

**(3)** [Section 5] We evaluate the performance of DCM on a range of synthetic datasets generated with various structural equation types for all three forms of causal queries. We find that DCM consistently outperforms existing state-of-the-art methods (Sánchez-Martín et al., 2022; Khemakhem et al., 2021). In fact, for certain interventional and counterfactual queries such as those arising with *nonadditive noise* models, DCM is better by an order of magnitude or more than these existing approaches. Additionally, we demonstrate the favorable performance of DCM on an interventional query experiment conducted on fMRI data.

**Related Work.** Over the years, a variety of methods have been developed in the causal inference literature for answering interventional and/or counterfactual queries including non-parametric methods (Shalit et al., 2017; Alaa & Van Der Schaar, 2017; Muandet et al., 2021) and probabilistic modeling methods (Zečević et al., 2021a). More relevant to our approach is a recent series of work, including (Moraffah et al., 2020; Pawlowski et al., 2020; Kocaoglu et al., 2018; Parafita & Vitrià, 2020; Zečević et al., 2021b; Garrido et al., 2021; Karimi et al., 2020; Sánchez-Martín et al., 2022; Khemakhem et al., 2021; Sanchez & Tsaftaris, 2022) that have demonstrated the success of using deep (conditional) generative models for this task.

Karimi et al. (2020) propose an approach for answering interventional queries by fitting a *conditional variational autoencoder* to each conditional in the Markov factorization implied by the causal graph. Also using the ideas of variational inference and normalizing flows, Pawlowski et al. (2020) propose schemes for counterfactual inference.

In Khemakhem et al. (2021), the authors propose an autoregressive normalizing flow for causal discovery and queries, referred to as *CAREFL*. CAREFL is also applicable even with only knowledge of the causal ordering rather than the full causal graph as we require. However, as also noted by Sánchez-Martín et al. (2022), when the causal graph is present, CAREFL is unable to exploit the absence of edges fully as it reduces a causal graph to its causal ordering (which may not be unique). Using normalizing flows for answering causal queries with causal DAG was also explored by (Balgi et al., 2022), however their approach does not have any theoretical guarantees on their counterfactual estimates, and their experimental evaluation is quite limited.

Sánchez-Martin et al. (2022) propose *VACA*, which uses graph neural networks (GNNs) in the form of a *variational graph autoencoder* to sample from the observational, interventional, and counterfactual distribution. VACA can utilize the inherent graph structure through the GNN, however, suffers in empirical performance (see Section 5). Furthermore, the usage of the GNN leads to undesirable design constraints, e.g., the encoder GNN cannot have hidden layers (Sánchez-Martin et al., 2022).

A very recent work by Javaloy et al. (2023) combines the idea of using autoregressive normalizing flows (as in CAREFL) and modeling the entire causal graph as one model (as in VACA), to reduce possible error propagation in the graph. The authors further generalize the theoretical results from Khemakhem et al. (2021) beyond affine autoregressive normalizing flows and establish a clearer connection to SCMs by providing a more direct way of applying the do-operator. In contrast to our work, Javaloy et al. (2023) focuses on modeling the whole causal graph as one model. Modeling on a per-node basis has several advantages over a single model in terms of flexibility and computational efficiency because it permits individual node models to be trained in parallel. For example, on a single experiment, we observed that our DCM approach trains roughly seven times faster than CAREFL and nine times faster than VACA (see Appendix D.1).

Sanchez & Tsaftaris (2022) use diffusion models for counterfactual estimation, focusing on the bivariate graph case with an image class causing an image. The authors train a diffusion model to generate images and use the abduction-action-prediction procedure from Pearl et al. (2016) as well as classifier guidance (Dhariwal & Nichol, 2021) to generate counterfactual images. However, this is solely for bivariate models and requires training a separate classifier for intermediate diffusion images, and exhibits poor performance for more complex images e.g., ImageNet (Deng et al., 2009). Our approach distinguishes itself from Sanchez & Tsaftaris (2022) as it can handle general causal graphs (beyond the simpler two node setting) and operates on continuous variables (beyond the simpler case of a discrete label and image). Our experimental evaluations are more general as it also covers interventional queries not addressed by Sanchez & Tsaftaris (2022). Finally, in terms of theoretical contribution, we provide rigorous conditions on the latent model and structural equations under which we can estimate counterfactuals. Even for two variable setting considered by Sanchez & Tsaftaris (2022) such a theoretical understanding was previously missing.

Finally, a diffusion model based approach has also been recently proposed for the different task of causal discovery under additive noise models, where the diffusion model serves as an approximation to the Hessian functions (Sanchez et al., 2022). This raises the interesting question of whether the diffusion models can be adapted for solving the end-to-end problem from discovery to inference.

## 2 Preliminaries

**Notation.** To distinguish random variables from their instantiation, we represent the former with capital letters and the latter with the corresponding lowercase letters. To distinguish between the nodes in the causal graph and diffusion random variables, we use subscripts to denote graph nodes. Let $[n] := \{1, \ldots, n\}$.

**Structural Causal Models.** Consider a directed acyclic graph (DAG) $\mathcal{G}$ with nodes $\{1, \ldots, K\}$ in a topologically sorted order, where a node $i$ is represented by a (random) variable $X_i$ in some generic space $\mathcal{X}_i \subset \mathbb{R}^{d_i}$. Let $\mathrm{pa}_i$ be the parents of node $i$ in $\mathcal{G}$ and let $X_{\mathrm{pa}_i} := \{X_j\}_{j \in \mathrm{pa}_i}$ be the variables of the parents of node $i$. A structural causal model $\mathcal{M}$ describes the relationship between an observed/endogenous node $i$ and its causal parents. Formally, an SCM $\mathcal{M} := (\mathbf{F}, p(\mathbf{U}))$ determines how a set of $K$ endogenous random variables $\mathbf{X} := \{X_1, \ldots, X_K\}$ is generated from a set of exogenous random variables $\mathbf{U} := \{U_1, \ldots, U_K\}$ with prior distribution $p(\mathbf{U})$ via a set of structural equations, $\mathbf{F} := (f_1, \ldots, f_K)$ where $X_i := f_i(X_{\mathrm{pa}_i}, U_i)$ for $i \in [K]$. Throughout this paper, we assume that the unobserved random variables are jointly independent (Markovian SCM), and the DAG $\mathcal{G}$ is the graph induced by $\mathcal{M}$. Every SCM $\mathcal{M}$ entails a unique joint observational distribution satisfying the causal Markov assumption: $p(\mathbf{X}) = \prod_{i=1}^{K} p(X_i \mid X_{\mathrm{pa}_i})$.

Structural causal models address Pearl's causal hierarchy (or "ladder of causation"), which consists of three "layers" of causal queries in increasing complexity (Pearl, 2009a): observational (or associational), interventional, and counterfactual. As an example, an interventional query can be formulated as "What will be the effect on the population $\mathbf{X}$, if a variable $X_i$ is assigned a fixed value $\gamma_i$?" The do-operator $\mathrm{do}(X_i := \gamma_i)$ represents the effect of setting variable $X_i$ to $\gamma_i$. Note that our proposed framework allows for more general sets of interventions as well, such as interventions on multiple variables denoted as $\mathrm{do}(X_{\mathcal{I}} := \gamma)$ (where $\mathcal{I} \subseteq [K]$, $X_{\mathcal{I}} := (X_i)_{i \in \mathcal{I}}$, $\gamma \in \mathbb{R}^{|\mathcal{I}|}$). An intervention operation, $\mathrm{do}(X_{\mathcal{I}} := \gamma)$, transforms the original joint distribution into an interventional distribution denoted

by $p(\mathbf{X} \mid \mathrm{do}(X_{\mathcal{I}} := \gamma))$. On the other hand, a counterfactual query can be framed as "What would have been the outcome of a particular factual sample $x^{\mathrm{F}} := (x_1^{\mathrm{F}}, \ldots, x_K^{\mathrm{F}})$, if the value of $X_{\mathcal{I}}$ had been set to $\gamma$?". Counterfactual estimation may be performed through the three-step procedure of 1) abduction: estimation of the exogenous noise $U$, 2) action: intervene $\mathrm{do}(X_{\mathcal{I}} := \gamma)$, and 3) prediction: estimating $x^{\mathrm{CF}}$ using the abducted noise and intervention values. Note that in Step 1, we perform deterministic counterfactual reasoning, focusing on counterfactuals pertaining to a single unit of the population.

**Diffusion Models.** Given data from distribution $X^0 \sim Q$, the objective of diffusion models is to construct an efficiently sampleable distribution approximating $Q$. Denoising diffusion probabilistic models (DDPMs) (Sohl-Dickstein et al., 2015; Ho et al., 2020) accomplish this by introducing a forward noising process that adds isotropic Gaussian noise at each time step and a learned reverse denoising process. A common representation of diffusion models is a fixed Markov chain that adds Gaussian noise with variances $\beta_1, \ldots, \beta_T \in (0, 1)$, generating latent variables $X^1, \ldots, X^T$, $q(X^t \mid x^{t-1}) = \mathcal{N}(X^t; \sqrt{1-\beta_t} x^{t-1}, \beta_t I)$ and $q(X^t \mid x^0) = \mathcal{N}(X^t; \sqrt{\alpha_t} x^0, (1-\alpha_t)I)$, where $\alpha_t := \prod_{i=1}^t (1-\beta_i)$. Here, $T \in \mathbb{Z}^+$, and $t \in \{0, \ldots, T\}$ denotes the time index.

By choosing sufficiently large $T$ and $\alpha_t$ that converge to 0, we have $X^T$ is distributed as an isotropic Gaussian distribution. The learned reverse diffusion process attempts to approximate the intractable $q(X^{t-1} \mid x^t)$ using a neural network and is defined as a Markov chain with Gaussian transitions, $p_\theta(X^{t-1} \mid x^t) = \mathcal{N}(X^{t-1}; \mu_\theta(x^t, t), \Sigma_\theta(x^t, t))$. Rather than predicting $\mu_\theta$ directly, the network could instead predict the Gaussian noise $\varepsilon$ from $x^t = \sqrt{\alpha_t} x^0 + \sqrt{1-\alpha_t}\varepsilon$. Ho et al. (2020) found that modeling $\varepsilon$ instead of $\mu_\theta$, fixing $\Sigma_\theta$, and using the following reweighted loss function

$$\mathbb{E}_{\substack{t \sim \mathrm{Unif}\{[T]\} \\ X^0 \sim Q \\ \varepsilon \sim \mathcal{N}(0,I)}} [\|\varepsilon - \varepsilon_\theta(\sqrt{\alpha_t} X^0 + \sqrt{1-\alpha_t}\varepsilon, t)\|^2], \tag{1}$$

works well empirically. We also utilize this loss function in our training.

Song et al. (2021) demonstrate that it is possible to take a pretrained standard denoising diffusion probabilistic model (DDPM) and generalize the generation to non-Markovian processes. In particular, it is possible to use a pretrained DDPM model to obtain a deterministic sample given noise $X^T$, known as the denoising diffusion implicit model (DDIM), with *reverse implicit diffusion process*

$$X^{t-1} := \sqrt{\frac{\alpha_{t-1}}{\alpha_t}} X^t - \varepsilon_\theta(X^t, t) \left( \sqrt{\alpha_{t-1}(1-\alpha_t)/\alpha_t} - \sqrt{1-\alpha_{t-1}} \right). \tag{2}$$

Note that the $X^t$ here is deterministic. We also use a *forward implicit diffusion process* introduced by Song et al. (2021), derived from rewriting the DDIM process Eq. 2 as an ordinary differential equation (ODE) and considering the Euler method approximation in the forward direction to obtain

$$X^{t+1} := \sqrt{\frac{\alpha_{t+1}}{\alpha_t}} X^t + \varepsilon_\theta(X^t, t) \left( \sqrt{1-\alpha_{t+1}} - \sqrt{\alpha_{t+1}(1-\alpha_t)/\alpha_t} \right). \tag{3}$$

We utilize the DDIM framework in this work, in particular Eqs. 3 and 2 will define the encoding (forward) and decoding (reverse) processes. Note that this ensures deterministic encoding and decoding. This construction produces a unique latent variable per observation, as well as a unique decoding, and also ensures we obtain the same output for repeated counterfactual queries.

# 3 DCMs: Diffusion-based Causal Models

In this section, we present our DCM approach for modeling the SCMs and to answer causal queries. The DCM approach falls in a general class of techniques that try to model a structural causal model by using an encoder-decoder pair. Consider a data generating process $X = f(X_{pa}, U)$. The goal will to construct an encoding function $g$ and a decoding function $h$. The encoding function $g$ attempts to represent the information in $U$: for a pair $(X, X_{pa})$, $Z := g(X, X_{pa})$ is the latent variable. The decoder takes the input $Z$ and $X_{pa}$ as input to attempt to reconstruct $X$: $\hat{X} = h(Z, X_{pa})$, where under perfect reconstruction, $\hat{X} = X$. The decoding function $h$ mimics the true structural

equation $f$, although it does not need to be exactly equal. For example, there are infinitely many encodings that satisfy $Z = r(U)$ for all $U$ for an invertible function $r$.

We first explain the construction and the training process of a DCM, and then explain how the model can be used for answering various causal queries. We start with some notations.

- Define $Z_i^t$ to be the $i$th endogenous node value at diffusion step $t$ of the *forward* implicit diffusion process (Eq. 3), and let $Z_i := Z_i^T$.

- Define $\hat{X}_i^t$ to be $i$th endogenous node value at diffusion step $t$ of the *reverse* implicit diffusion process (Eq. 2), and let $\hat{X}_i := \hat{X}_i^0$.

**Training a DCM.** We train a diffusion model for *each node*, taking denoised parent values as input. The parent values can be interpreted as additional covariates to the model, where one may choose to use classifier free guidance to incorporate the covariates (Ho & Salimans, 2021). Empirically, we find that simply concatenating the covariates results in better performance than classifier free guidance.

We use the $\varepsilon_\theta$ parametrization for the diffusion model from Ho et al. (2020), representing the diffusion model for node $i$ as $\varepsilon_\theta^i(X, X_{\mathrm{pa}_i}, t)$. The complete training procedure presented in Algorithm 1 is only slightly modified from the usual training procedure, with the additions of the parents as covariates and training a diffusion model for each node. Since the generative models learned for generation of different endogenous nodes do not affect training of each other, these models may be trained in parallel. For each node in the graph, we can train a model in parallel as each diffusion model only requires the current node and parent values. Our final DCM model is just a combination of these $K$ trained diffusion models $\varepsilon_\theta^1, \ldots, \varepsilon_\theta^K$.

---

**Algorithm 1** DCM Training

**Input:** Distribution $Q$, scale factors $\{\alpha_t\}_{t=1}^T$, causal DAG $\mathcal{G}$ with node $i$ represented by $X_i$

1: **while** not converged **do**
2:     Sample $X^0 \sim Q$
3:     **for** $i = 1, \ldots, K$ **do**
4:         $t \sim \mathrm{Unif}[\{1, \ldots, T\}]$
5:         $\varepsilon \sim \mathcal{N}(0, I_{d_i})$                                                       $\{d_i$ is the dimension of $X_i\}$
6:         Update parameters of node $i$'s diffusion model $\varepsilon_\theta^i$, by minimizing the following loss: $\|\varepsilon - \varepsilon_\theta^i(\sqrt{\alpha_t}X_i^0 + \sqrt{1-\alpha_t}\varepsilon, X_{\mathrm{pa}_i}^0, t)\|_2^2$    (based on Eq. 1)
7:     **end for**
8: **end while**

---

We use all the variables $(X_1, \ldots, X_K)$ for the training procedure, because a priori, we do not assume anything on the possible causal queries, i.e., we allow for all possible target variables, intervened variables, etc. However, if we are only interested in some pre-defined set of queries, then the graph could be reduced accordingly. For example, if we are only interested in counterfactual estimate of a particular node with respect to an intervention of a predecessor, one can simply reduce it to a subgraph containing the target node, the intervened node and a backdoor adjustment set (e.g., the ancestors of the intervened node). This then reduces to only learning a single diffusion model.

One major advantage of our proposed DCM approach is the ability to generalize to larger graphs. Since each diffusion model only uses the parents as input, modeling each node depends only on the incoming degree of the node (the number of causal parents). While the number of diffusion models scales with the number of non-root nodes, each model is generally small in terms of its parameter size and can be trained in parallel. Additionally, we may apply the proposed DCM approach to any setting where diffusion models are applicable: continuous variables, high dimensional settings, categorical data, images, etc.

**Encoding and Decoding Steps with DCM.** With a DCM, the encoding (resp. decoding) process is identical to the DDIM encoding (resp. decoding) process except we include the parent values as additional covariates. Note that, given the model $\epsilon_\theta$, DDIM is a deterministic process as laid out in Eqs. 2 and 3. Let us focus on a node $i \in [K]$ (same process is repeated for each node $i$). The encoding process takes $X_i$ and its parent values $X_{\mathrm{pa}_i}$ as input and maps it to a latent variable $Z_i$. The decoding process takes $Z_i$ and $X_{\mathrm{pa}_i}$ as input to construct $\hat{X}_i$ (an approximation of $X_i$). Formally, using the forward implicit diffusion process in Eq. 3, given a sample $X_i$, we encode a unique latent variable

$Z_i := Z_i^T$, using the recursive formula

$$Z_i^{t+1} := \sqrt{\frac{\alpha_{t+1}}{\alpha_t}} Z_i^t + \varepsilon_\theta^i(Z_i^t, X_{\mathrm{pa}_i}, t) \left( \sqrt{1 - \alpha_{t+1}} - \sqrt{\frac{\alpha_{t+1}(1 - \alpha_t)}{\alpha_t}} \right), \forall t = 0, .., T - 1, \qquad (4)$$

where $Z_i^0 := X_i$. The latent variable $Z_i$ acts as a proxy for the exogenous noise $U_i$. Using the reverse implicit diffusion process from DDIM in Eq. 2, given a latent vector $Z_i$ we obtain a deterministic decoding $\hat{X}_i := \hat{X}_i^0$, using the recursive formula

$$\hat{X}_i^{t-1} := \sqrt{\frac{\alpha_{t-1}}{\alpha_t}} \hat{X}_i^t - \varepsilon_\theta^i(\hat{X}_i^t, X_{\mathrm{pa}_i}, t) \left( \sqrt{\frac{\alpha_{t-1}(1 - \alpha_t)}{\alpha_t}} - \sqrt{1 - \alpha_{t-1}} \right), \text{ for all } t = T, \ldots, 1, \qquad (5)$$

where $\hat{X}^T := Z_i$. In the following, we use $\mathsf{Enc}_i(X_i, X_{\mathrm{pa}_i})$ and $\mathsf{Dec}_i(Z_i, X_{\mathrm{pa}_i})$ to denote the encoding and decoding functions for node $i$ defined in Eqns. 4 and 5 respectively. See Algorithms 2 and 3 for detailed pseudocodes.

---

**Algorithm 2** $\mathsf{Enc}_i(X_i, X_{\mathrm{pa}_i})$

---

**Input:** $X_i, X_{\mathrm{pa}_i}$

1: $Z_i^0 \leftarrow X_i$
2: **for** $t = 0, \ldots, T - 1$ **do**
3: $\quad Z_i^{t+1} \leftarrow \sqrt{\frac{\alpha_{t+1}}{\alpha_t}} Z_i^t + \varepsilon_\theta^i(Z_i^t, X_{\mathrm{pa}_i}, t) \left( \sqrt{1 - \alpha_{t+1}} - \sqrt{\frac{\alpha_{t+1}(1 - \alpha_t)}{\alpha_t}} \right)$
4: **end for**
5: Return $Z_i := Z_i^T$

---

**Algorithm 3** $\mathsf{Dec}_i(Z_i, X_{\mathrm{pa}_i})$

---

**Input:** $Z_i, X_{\mathrm{pa}_i}$

1: $\hat{X}^T \leftarrow Z_i$
2: **for** $t = T, \ldots, 1$ **do**
3: $\quad \hat{X}_i^{t-1} \leftarrow \sqrt{\frac{\alpha_{t-1}}{\alpha_t}} \hat{X}_i^t - \varepsilon_\theta^i(\hat{X}_i^t, X_{\mathrm{pa}_i}, t) \left( \sqrt{\frac{\alpha_{t-1}(1 - \alpha_t)}{\alpha_t}} - \sqrt{1 - \alpha_{t-1}} \right)$
4: **end for**
5: Return $\hat{X}_i := \hat{X}_i^0$

---

**Answering Causal Queries with a Trained DCM.** We now describe how a trained DCM model can be used for (approximately) answering causal queries. Answering observational and interventional queries require sampling from the observational and the interventional distribution respectively. With counterfactuals, a query is at the unit level, where the structural assignments are changed, but the exogenous noise is identical to that of the observed datum.

**(a) Generating Samples for Observational/Interventional Queries.** Samples from a DCM model that approximates the interventional distribution $p(\mathbf{X} \mid \mathrm{do}(X_{\mathcal{I}} := \gamma))$ can be generated as follows. For an intervened node $i$ with intervention $\gamma_i$, the sampled value is always the intervention value, therefore we generate $\hat{X}_i := \gamma_i$. For a non-intervened node $i$, assume by induction we have the generated parent values $\hat{X}_{\mathrm{pa}_i}$. To generate $\hat{X}_i$, we first sample the latent vector $Z_i \sim \mathcal{N}(0, I_{d_i})$ where $d_i$ is the dimension of $X_i$. Then taking $Z_i$ as the noise for node $i$, we compute $\hat{X}_i := \mathsf{Dec}_i(Z_i, \hat{X}_{\mathrm{pa}_i})$ as the generated sample value for node $i$. This value $\hat{X}_i$ is then used as the parent value for the children of node $i$. Samples from a DCM model that approximates the observational distribution $p(\mathbf{X})$ can be generated by setting $\mathcal{I} = \emptyset$. See Algorithm 4 for the pseudocode.

**(b) Counterfactual Queries.** Consider a factual observation $x^{\mathrm{F}} := (x_1^{\mathrm{F}}, \ldots, x_K^{\mathrm{F}})$ and interventions on a set of nodes $\mathcal{I}$ with values $\gamma$. We use a DCM model to construct a counterfactual estimate $\hat{x}^{\mathrm{CF}}$ as follows. The counterfactual estimate only differs from the factual value on intervened nodes or descendants of an intervened node. Similarly to interventional queries, for each intervened node $i \in \mathcal{I}$, $\hat{x}_i^{\mathrm{CF}} := \gamma_i$. For each non-intervened node $i$ that is a descendant of any intervened node, assume by induction that we have the generated counterfactual estimates $\hat{x}_{\mathrm{pa}_i}^{\mathrm{CF}}$. To obtain $\hat{x}_i^{\mathrm{CF}}$,

---

**Algorithm 4** Observational/Interventional Sampling

---

**Input:** Intervention set $\mathcal{I}$ with values $\gamma$ ($I = \emptyset$ for observational sampling)

1: **for** $i = 1, \ldots, K$ **do**           {in topological order}
2:     $Z_i \sim \mathcal{N}(0, I_{d_i})$
3:     **if** $i \in \mathcal{I}$ **then**
4:        $\hat{X}_i \leftarrow \gamma_i$
5:     **else**
6:        $\hat{X}_i \leftarrow \mathsf{Dec}_i(Z_i, \hat{X}_{\mathrm{pa}_i})$
7:     **end if**
8: **end for**
9: Return $\hat{X} := (\hat{X}_1, \ldots, \hat{X}_K)$

---

**Algorithm 5** Counterfactual Estimation

---

**Input:** Intervention set $\mathcal{I}$ with values $\gamma$, factual sample $x^{\mathrm{F}} := (x_1^{\mathrm{F}}, \ldots, x_K^{\mathrm{F}})$

1: **for** $i = 1, \ldots, K$ **do**           {in topological order}
2:     **if** $i \in \mathcal{I}$ **then**
3:        $\hat{x}_i^{\mathrm{CF}} \leftarrow \gamma_i$
4:     **else if** $i$ is not a descendant of any intervened node in $\mathcal{I}$ **then**
5:        $\hat{x}_i^{\mathrm{CF}} \leftarrow x_i^{\mathrm{F}}$
6:     **else**
7:        $z_i^{\mathrm{F}} \leftarrow \mathsf{Enc}_i(x_i^{\mathrm{F}}, x_{\mathrm{pa}_i}^{\mathrm{F}})$           {abduction step}
8:        $\hat{x}_i^{\mathrm{CF}} \leftarrow \mathsf{Dec}_i(z_i^{\mathrm{F}}, \hat{x}_{\mathrm{pa}_i})$           {action and prediction steps}
9:     **end if**
10: **end for**
11: Return $\hat{x}^{\mathrm{CF}} := (\hat{x}_1^{\mathrm{CF}}, \ldots, \hat{x}_K^{\mathrm{CF}})$

---

we first define the estimated factual noise as $\hat{z}_i^{\mathrm{F}} := \mathsf{Enc}_i(x_i^{\mathrm{F}}, x_{\mathrm{pa}_i}^{\mathrm{F}})$. Then we generate our counterfactual estimate by using $\hat{z}_i^{\mathrm{F}}$ as the noise for node $i$, by decoding, $\hat{x}_i^{\mathrm{CF}} := \mathsf{Dec}_i(\hat{z}_i^{\mathrm{F}}, \hat{x}_{\mathrm{pa}_i}^{\mathrm{CF}})$. See Algorithm 5 for the pseudocode.

Note that with $x^{\mathrm{F}}$, we assumed full observability,[1] since because Algorithm 5 produces a counterfactual estimate for each node. However, when intervening on $X_{\mathcal{I}}$ and if the only quantity of interest is counterfactual on some $X^{\star}$, then you only need factual samples from $\{X_i : X_i \text{ is on a path from } X_{\mathcal{I}} \to X^{\star}\}$ (Saha & Garain, 2022). In practice, this could be further relaxed by imputing for missing data, which is beyond the scope of this work.

## 4 Bounding Counterfactual Error

We now establish *sufficient* conditions under which the counterfactual estimation error can be bounded. In fact, the results in this section not only hold for diffusion models, but to a more general setting of conditional latent variable models satisfying certain properties. Another feature of this result, is that it also extends, under an additional assumption, to the more challenging higher-dimensional case. All proofs from this section are collected in Appendix A.

We focus on learning a single conditional latent variable model for an endogenous node $X_i$, given its parents $X_{\mathrm{pa}_i}$, as the models learned for different endogenous nodes do not affect each other. Since the choice of node $i$ plays no role, we drop the subscript $i$ in the following and refer to the node of interest as $X$, its causal parents as $X_{\mathrm{pa}}$, its corresponding exogenous variables as $U$, and its structural equation as $X := f(X_{\mathrm{pa}}, U)$. Let the encoding function $g : \mathcal{X} \times \mathcal{X}_{\mathrm{pa}} \to \mathcal{Z}$ and the decoding function $h : \mathcal{Z} \times \mathcal{X}_{\mathrm{pa}} \to \mathcal{X}$, where $\mathcal{Z}$ is the latent space. In the DCM context, the functions $g$ and $h$ correspond to Enc and Dec functions, respectively.

It is well-known that certain counterfactual queries are not identifiable from observational data without making assumptions on the functional relationships, even under causal sufficiency (Pearl, 2009b). Consequently, recent research has been directed towards understanding the conditions under which identifiability results can be obtained (Lu et al.,

---

[1]This is a common assumption in literature also made in all the related work e.g., (Sánchez-Martin et al., 2022; Khemakhem et al., 2021; Pawlowski et al., 2020).

2020; Nasr-Esfahany & Kiciman, 2023; Nasr-Esfahany et al., 2023). Assumption 2 of our Theorem 1, ensures that the true counterfactual outcome is identifiable, see e.g., (Lu et al., 2020, Theorem 1) or (Nasr-Esfahany & Kiciman, 2023, Theorem 5). In the context of learned structural causal models to determine whether a given counterfactual query can be answered with sufficient accuracy, requires also assumptions on the learned SCM, e.g., encoder and decoder in this case.

Our first result presents sufficient conditions on the latent variable encoding function and the structural equation under which we can recover latent exogenous variable up to a (possibly nonlinear) invertible function. We start with a one-dimensional exogenous noise $U$ and variable $X \in \mathcal{X} \subset \mathbb{R}$. In Section 4.1, we provide a similar theorem for the higher-dimensional case where $X \in \mathbb{R}^m$ for $m \geq 3$ in Theorem 2 with a stronger assumption on the Jacobian of $f$ and $g$.

**Theorem 1.** *Assume for $X \in \mathcal{X} \subset \mathbb{R}$ and exogenous noise $U \sim \mathrm{Unif}[0,1]$, $X$ satisfies the structural equation: $X := f(X_{\mathrm{pa}}, U)$, where $X_{\mathrm{pa}} \in \mathcal{X}_{\mathrm{pa}} \subset \mathbb{R}^d$ are the parents of node $X$ and $U \perp\!\!\!\perp X_{\mathrm{pa}}$. Consider an encoder-decoder model with encoding function $g : \mathcal{X} \times \mathcal{X}_{\mathrm{pa}} \to \mathcal{Z}$ and decoding function $h : \mathcal{Z} \times \mathcal{X}_{\mathrm{pa}} \to \mathcal{X}$, $Z := g(X, X_{\mathrm{pa}}), \quad \hat{X} := h(Z, X_{\mathrm{pa}})$. Assume the following conditions:*

1. *The encoding is independent of the parent values, $g(X, X_{\mathrm{pa}}) \perp\!\!\!\perp X_{\mathrm{pa}}$.*

2. *The structural equation $f$ is differentiable and strictly increasing with respect to $U$ for all $x_{\mathrm{pa}} \in \mathcal{X}_{\mathrm{pa}}$.*

3. *The encoding $g$ is invertible and differentiable with respect to $X$ for all $x_{\mathrm{pa}} \in \mathcal{X}_{\mathrm{pa}}$.*

*Then, $g(X, X_{\mathrm{pa}}) = \tilde{q}(U)$ for an invertible function $\tilde{q}$.*

**Discussion on Assumptions Underlying Theorem 1.**

**(1)** Assumption 1 of independence between the encoding and the parent values may appear strong, but is in fact often valid. For example, in the additive noise setting with $f(X_{\mathrm{pa}}, U) := f'(X_{\mathrm{pa}}) + U$ where $X_{\mathrm{pa}}$ and $U$ are independent, if the fitted model $\hat{f} \equiv f'$, then the encoder $g(X, X_{\mathrm{pa}}) = f(X_{\mathrm{pa}}, U) - \hat{f}(X_{\mathrm{pa}}) = U$ and by definition $U$ is independent of $X_{\mathrm{pa}}$.[2] The same assumption also appears in other related results on counterfactual identifiability in bijective SCMs, see, e.g., (Nasr-Esfahany et al., 2023, Theorem 5.3) and proof of Theorem 5 in (Nasr-Esfahany & Kiciman, 2023). We conduct empirical tests to further confirm this assumption by examining the dependence between the parents and encoding values. Our experimental results show that DCMs consistently fail to reject the null hypothesis of independence. This implies that independent encodings can be found in practice. We provide the details of these experiments in Appendix B.[3]

**(2)** Assumption 2 is always satisfied under the additive noise model, where $f(X_{\mathrm{pa}}, U) := f'(X_{\mathrm{pa}}) + U$. It is also satisfied by post non-linear models, under the standard requirements on identifiability as expressed in (Zhang & Hyvarinen, 2012).[4] Assumption 2 is also satisfied by heteroscedastic noise models (Strobl & Lasko, 2023).[5] Again, the recent results about counterfactual identifiability, e.g., (Nasr-Esfahany & Kiciman, 2023, Theorem 5) and (Lu et al., 2020, Theorem 1), also utilize the same assumption.

Through our strictly increasing in $U$ assumption, we obviate distinguishing cases like $f(X_{\mathrm{pa}}, U) := X_{\mathrm{pa}} + U$ vs. $f(X_{\mathrm{pa}}, U) := X_{\mathrm{pa}} - U$, which otherwise will be indistinguishable for symmetric distributions $U$ (see also remark below). For example, for any fixed value of $X_{\mathrm{pa}}$, $f(X_{\mathrm{pa}}, U) := X_{\mathrm{pa}} - U$ is not increasing in $U$, so such structural equations are automatically eliminated in Theorem 1. In other words, among these two cases, this assumption only permits the additive noise form $f(X_{\mathrm{pa}}, U) := X_{\mathrm{pa}} + U$.

---

[2]In general, if we have a good approximation $f$ by some $\hat{f}$, then the encoding $g(X, X_{\mathrm{pa}}) = X - \hat{f}(X_{\mathrm{pa}})$ would be close to $U$, as also noted by (Hoyer et al., 2008).

[3]To encourage independence, one could also modify the original diffusion model training objective to add a Hilbert-Schmidt independence criterion (HSIC) (Gretton et al., 2007) regularization term. Our experiments did not show a clear benefit of using this modified objective, and we leave further investigation here for future work.

[4]Zhang & Hyvarinen (2012) (see Eqn. 2) defined post-nonlinear models as $f(X_{\mathrm{pa}}, U) := f_2(f_1(X_{\mathrm{pa}}) + U)$ where $X_{\mathrm{pa}}$ and $U$ are independent, function $f_1$ is nonconstant, and $f_2$ is invertible. The invertibility assumption on $f_2$ implies that $f$ is strictly increasing (or decreasing) in $U$. Additionally, (Zhang & Hyvarinen, 2012) make a differentiability of $f_2$ assumption (see Assumption A1) for identifiability, which implies the $f$ is differentiable in $U$.

[5]Heteroscedastic noise models are generally defined as $f(X_{\mathrm{pa}}, U) := f'(X_{pa}) + g(X_{pa}) \cdot U$ where the function $g$ is assumed to be strictly positive (see Definition 1 in (Strobl & Lasko, 2023), which makes it compatible with our Assumption 2 of Theorem 1.

**(3)** We may consider transformations of the uniform noise $U$ to obtain other settings, for example additive Gaussian noise. For a continuous random variable $U'$ with invertible CDF $F$ and the structural equation $f(\cdot, F(\cdot))$, we have $U' \overset{d}{=} F^{-1}(U)$ and the results similarly hold.

We now discuss some consequences of Theorem 1 for estimating counterfactual outcomes.

**1. Perfect Estimation.** Using Theorem 1, we now look at a condition under which the counterfactual estimate produced by the encoder-decoder model matches the true counterfactual outcome.[6]. The idea here been, if no information is lost in the encoding and decoding steps, i.e., $h(g(X, X_{\mathrm{pa}}), X_{\mathrm{pa}}) = X$, and assuming Theorem 1 $(g(X, X_{\mathrm{pa}}) = \tilde{q}(U))$, we have $h(\tilde{q}(U), X_{\mathrm{pa}}) = X = f(X_{\mathrm{pa}}, \tilde{q}^{-1}(U))$. This means that in the abduction step, the encoder-decoder model could recover $\tilde{q}(U)$, but in the prediction step, it first applies the inverse of $\tilde{q}$ to the recovered exogenous variable, and then $f$. Thus, the counterfactual estimate equals the true counterfactual outcome. We formalize this in Corollary 1.

**Corollary 1.** *Assume the conditions of Theorem 1. Furthermore, assume the encoder-decoder model pair $(g, h)$ satisfies: $h(g(X, X_{\mathrm{pa}}), X_{\mathrm{pa}}) = X$. Consider a factual sample pair $x^{\mathrm{F}} := (x, x_{\mathrm{pa}})$ where $x := f(x_{\mathrm{pa}}, u)$ and an intervention $\mathrm{do}(X_{\mathrm{pa}} := \gamma)$. Then, the counterfactual estimate, given by $h(g(x, x_{\mathrm{pa}}), \gamma)$ matches the true counterfactual outcome $x^{\mathrm{CF}} := f(\gamma, u)$.*

**Comparison with Recent Related Work.** Recent studies by Nasr-Esfahany & Kiciman (2023) and Nasr-Esfahany et al. (2023) have explored the problem of estimating counterfactual outcomes with learned SCMs. In particular, Nasr-Esfahany & Kiciman (2023, Theorem 5) consider a setting where the SCM $X := f(X_{\mathrm{pa}}, U)$ is learned with a *bijective* (deep conditional generative) model $\hat{f}(X_{\mathrm{pa}}, \hat{U})$. Nasr-Esfahany et al. (2023, Theorem 5.3) considered a closely related problem of learning a ground-truth bijective SCM. The conditions underlying ours and these results are not directly comparable because, unlike our setup, they do not explicitly consider an encoder-decoder model. Our results provide precise conditions on the encoder $g$ and decoder $h$ for recovering the correct counterfactual outcome, and we can extend these results to obtain counterfactual estimation error bounds under relaxed assumptions, a problem that has not been addressed previously.

Counterfactual identifiability in a different context of reinforcement learning was also established by (Lu et al., 2020). Their result relies on incomparable assumptions on state-action pairs. Furthermore, our proof techniques are quite different from Lu et al. (2020) who rely on a technique based on analyzing conditional quantile, unlike an algebraic technique employed here.

**2. Estimation Error.** Another consequence of Theorem 1 is that it can bound the counterfactual error in terms of the reconstruction error of the encoder-decoder model. Informally, the following corollary shows that if the reconstruction $h(g(X, X_{\mathrm{pa}}), X_{\mathrm{pa}})$ is "close" to $X$ (measured under some metric $d(\cdot, \cdot)$), then such encoder-decoder models can provide "good" counterfactual estimates. To the best of our knowledge, this is the first result that establishes a bound on the counterfactual error in relation to the reconstruction error of these encoder-decoder models.

**Corollary 2.** *Let $\gamma \geq 0$. Assume the conditions of Theorem 1. Furthermore, assume the encoder-decoder model pair $(g, h)$ under some metric $d$ (e.g., $\| \cdot \|_2$), has reconstruction error less than $\tau$: $d(h(g(X, X_{\mathrm{pa}}), X_{\mathrm{pa}}), X) \leq \tau$. Consider a factual sample pair $x^{\mathrm{F}} := (x, x_{\mathrm{pa}})$ where $x := f(x_{\mathrm{pa}}, u)$ and an intervention $\mathrm{do}(X_{\mathrm{pa}} := \gamma)$. Then, the error between the true counterfactual $x^{\mathrm{CF}} := f(\gamma, u)$ and counterfactual estimate given by $h(g(x, x_{\mathrm{pa}}), \gamma)$ is at most $\tau$. In other words, $d(h(g(x, x_{\mathrm{pa}}), \gamma), x^{\mathrm{CF}}) \leq \tau$.*

The above result suggests that the reconstruction error can serve as an estimate for the counterfactual error. While the true value of $\tau$ is unknown, we may compute a reasonable bound by computing the reconstruction error over the dataset.

## 4.1 Extension of Theorem 1 to Higher-Dimensional Setting

In this section, we present an extension of Theorem 1 to a higher dimensional setting and use it to provide counterfactual identifiability and estimation error results. Since we are now dealing with vector-valued functions, we use Jacobians. Let $Jf_{x_{\mathrm{pa}}}$ denote the Jacobian matrix obtained by evaluating the Jacobian (with respect to $U$) of $f(X_{\mathrm{pa}}, U)$ at $X_{\mathrm{pa}} = x_{\mathrm{pa}}$. Let $Jg_{x_{\mathrm{pa}}}$ denote the Jacobian matrix obtained by evaluating the Jacobian (with respect to $X$) of $g(X, X_{\mathrm{pa}})$ at $X_{\mathrm{pa}} = x_{\mathrm{pa}}$.

---

[6]Note that identifiability of the counterfactual outcomes, does not require identifiability of the SCM.

**Theorem 2.** *Assume for $X \in \mathcal{X} \subset \mathbb{R}^m$ and continuous exogenous noise $U \sim \mathrm{Unif}[0,1]^m$ for $m \geq 3$, and $X$ satisfies the structural equation*

$$X = f(X_{\mathrm{pa}}, U) \tag{6}$$

*where $X_{\mathrm{pa}} \in \mathcal{X}_{\mathrm{pa}} \subset \mathbb{R}^d$ are the parents of node $X$ and $U \perp\!\!\!\perp X_{\mathrm{pa}}$. Consider an encoder-decoder model with encoding function $g : \mathcal{X} \times \mathcal{X}_{\mathrm{pa}} \to \mathcal{Z}$ and decoding function $h : \mathcal{Z} \times \mathcal{X}_{\mathrm{pa}} \to \mathcal{X}$,*

$$Z := g(X, X_{\mathrm{pa}}), \quad \hat{X} := h(Z, X_{\mathrm{pa}}). \tag{7}$$

*Assume the following conditions:*

1. *The encoding is independent of the parent values, $g(X, X_{\mathrm{pa}}) \perp\!\!\!\perp X_{\mathrm{pa}}$.*

2. *The structural equation $f$ is invertible and differentiable with respect to $U$, and $Jf_{x_{\mathrm{pa}}}$ is p.d. for all $x_{\mathrm{pa}} \in \mathcal{X}_{\mathrm{pa}}$.*

3. *The encoding $g$ is invertible and differentiable with respect to $X$, and $Jg_{x_{\mathrm{pa}}}$ is p.d. for all $x_{\mathrm{pa}} \in \mathcal{X}_{\mathrm{pa}}$.*

4. *The encoding $q_{x_{\mathrm{pa}}}(U) := g(f(x_{\mathrm{pa}}, U), x_{\mathrm{pa}})$ satisfies $Jq_{x_{\mathrm{pa}}}|_{q_{x_{\mathrm{pa}}}^{-1}(z)} = c(x_{\mathrm{pa}})A$ for all $z \in \mathcal{Z}$ and $x_{\mathrm{pa}} \in \mathcal{X}_{\mathrm{pa}}$, where $c$ is a scalar function and $A$ is an orthogonal matrix.*

*Then, $g(f(X_{\mathrm{pa}}, U), X_{\mathrm{pa}}) = \tilde{q}(U)$ for an invertible function $\tilde{q}$.*

The interpretations behind Assumptions 1, 2, and 3 in Theorem 2 are similar to corresponding assumptions in Theorem 1. Assumption 4 however is technical, and we will explain the need for it below. In Corollaries 3 and 4 (Appendix A), we restate Corollaries 1 and 2 to this higher-dimensional setting. The proofs are identical to that of Corollaries 1 and 2, with the only change being that the role of Theorem 1 in those proofs is now replaced by Theorem 2.

**On Negative Result of Nasr-Esfahany & Kiciman (2023).** Nasr-Esfahany & Kiciman (2023) presented a general counterfactual impossibility identification result under multidimensional exogenous noise. The construction in Nasr-Esfahany & Kiciman (2023) considers two structural equations $f, f'$ that are indistinguishable in distribution. Formally, let $R \in \mathbb{R}^{m \times m}$ be a rotation matrix, and $U \in \mathbb{R}^m$ be a standard (isotropic) Gaussian random vector. Define,

$$f'(X_{\mathrm{pa}}, U) = \begin{cases} f(X_{\mathrm{pa}}, U) & \text{for } X_{\mathrm{pa}} \in A \\ f(X_{\mathrm{pa}}, R \cdot U) & \text{for } X_{\mathrm{pa}} \in B \end{cases}, \tag{8}$$

where the domain $\mathcal{X}_{\mathrm{pa}}$ is split into disjoint $A$ and $B$. Now, $f$ and $f'$ generate different counterfactual outcomes, for counterfactual queries with evidence in $A$ and intervention in $B$ (or the other way around).

In Theorem 2, we avoid this impossibility result by assuming that we can construct an encoding of a "special" kind captured through our Assumption 4. In particular, consider the encoding $q_{x_{\mathrm{pa}}}$ at a specific parent value $x_{\mathrm{pa}}$ as a function of the exogenous noise $U$. The assumption states that the Jacobian of the encoding is equal to $c(x_{\mathrm{pa}})A$ for a scalar function $c$ and orthogonal matrix $A$. However, it is important to acknowledge that this assumption is highly restrictive and difficult to verify, not to mention challenging to construct in practice with just observational data. Our intention is that these initial ideas can serve as a starting point for addressing the impossibility result, with the expectation that subsequent results will further refine and expand upon these ideas.

## 5 Experimental Evaluation

In this section, we evaluate the empirical performance of DCM for answering causal queries on both synthetic and real world data. Our primary objective through these experiments is not to tackle a specific causal inference problem, but to demonstrate that our DCM approach provides a unified and flexible framework for answering a wide range of observational, interventional, and counterfactual queries. For the real data experiments, we consider an interventional data experiment on fMRI data (Section 5.2) and an experimental study for the problem of estimating individual treatment effects (Appendix E).[7]

---

[7]Code for reproducibility: https://github.com/patrickrchao/DiffusionBasedCausalModels

**Diffusion Model Implementation and Training.** For our implementation of the $\varepsilon_\theta$ model in DCM, we use a simple fully connected neural network with three hidden layers of size $[128, 256, 256]$ and SiLU activation (Elfwing et al., 2018). We fit the model using Adam with a learning rate of 1e-4, batch size of 64, and train for 500 epochs. Since the root nodes lack parents, the only form of counterfactual reasoning involves directly intervening on the root node, which may be done trivially. Therefore, for root nodes, we do not train diffusion models and instead sample from the empirical distribution of the training data. Additional details about the diffusion model parameters are in Appendix C.1.

**Compared Approaches.** For a fair comparison, our evaluation centers on methodologies that allow for both interventional and counterfactual estimation. With this criteria, we primarily compare DCM to two recently proposed state-of-the-art schemes VACA (Sánchez-Martin et al., 2022) and CAREFL (Khemakhem et al., 2021), and a general regression model that assumes an additive noise model which we refer to as ANM.[8] For VACA and CAREFL, we use the code provided by their respective authors. The ANM approach performs model selection over a variety of models, including linear and gradient boosted regressor, and we use the implementation from the popular *DoWhy* causal inference package (Sharma et al., 2019; Blöbaum et al., 2022). Additional details on how ANM answers causal queries are provided in Appendix C.2 and implementation details for VACA, CAREFL, and ANM are in Appendix C.1.

## 5.1 Synthetic Data Experiments

For generating quantitative results, we use synthetic experiments since we know the exact structural equations, and hence we have access to the ground-truth observational, interventional, and counterfactual distributions. We present results on two larger graphs here, and defer results on a set of four smaller graphs to Appendix D.2. In Appendix D.3, we also present results by generating synthetic data based on the real world graph from the Sachs dataset (Sachs et al., 2005).

The two graphs, considered here, include a *ladder* graph structure (see Figure 9) and a randomly generated graph structure. Both graphs are comprised of 10 nodes of three dimensions each, and the random graph is a randomly sampled directed acyclic graph (see Appendix C.3 for more details). Since each diffusion model only uses the parents as input, modeling each node depends only on the incoming degree of the node (the number of causal parents).

Following Sánchez-Martin et al. (2022), for the observational and interventional distributions, we report the Maximum Mean Discrepancy (MMD) (Gretton et al., 2012) between the true and estimated distributions. For counterfactual estimation, we report the mean squared error (MSE) of the true and estimated counterfactual values. Again following Sánchez-Martin et al. (2022), we consider two broad classes of structural equations:

1. Additive Noise Model (NLIN): $f_i(X_{\mathrm{pa}_i}, U_i) = f'(X_{\mathrm{pa}_i}) + U_i$. In particular, we will be interested in the case where $f_i$'s are non-linear.
2. Nonadditive Noise Model (NADD): $f_i(X_{\mathrm{pa}_i}, U_i)$ is an arbitrary function of $X_{\mathrm{pa}_i}$ and $U_i$.

To prevent overfitting of hyperparameters, we randomly generate these structural equations for each initialization. Each structural equation is comprised a neural network with a single hidden layer of 16 units and SiLU activation (Elfwing et al., 2018) with random weights from sampled from $[-1, 1]$.

Each simulation generates $n = 5000$ samples as training data. Let $\hat{M}$ be a fitted causal model and $M^\star$ be the true causal model, both capable of generating observational and interventional samples, and answering counterfactual queries. Each pair of graphs and structural equation type is evaluated for 20 different initialization, and we report the mean value. We provide additional details about our observational, interventional, and counterfactual evaluation frameworks in Appendix C.4.

**Synthetic Experiments Results.** In Table 1, we provide the performance of all evaluated models for observational, interventional, and counterfactual queries, averaged over 20 separate initializations of models and training data, with the lowest value in each row bolded. The values are multiplied by 100 for clarity. We also provide boxplots of the performances in Figure 1 (Appendix D).

We see DCM and ANM are the most competitive approaches, with similar performance on observational and interventional queries. If the ANM is the correctly specified model, then the ANM encoding should be close to the true

---

[8]In spite of our best efforts, we were unable to run a proper comparison against the very recent approach proposed by Javaloy et al. (2023) due to challenges in adapting their code into our settings.

|  | | Metric | DCM $(\times 10^{-2})$ | ANM $(\times 10^{-2})$ | VACA $(\times 10^{-2})$ | CAREFL $(\times 10^{-2})$ |
|---|---|---|---|---|---|---|
| SCM | | | | | | |
| *Ladder* | NLIN | Obs. MMD | **0.44±0.14** | 0.63±0.21 | 2.82±0.83 | 13.41±1.14 |
| | | Int. MMD | **1.63±0.20** | 1.80±0.17 | 4.48±0.78 | 15.01±1.23 |
| | | CF. MSE | **3.42±1.67** | 10.65±2.48 | 41.03±19.00 | 17.46±6.04 |
| | NADD | Obs. MMD | **0.32±0.11** | 0.40±0.16 | 3.22±1.05 | 14.60±1.34 |
| | | Int. MMD | **1.54±0.17** | 1.57±0.15 | 5.13±1.16 | 16.87±1.85 |
| | | CF. MSE | **4.28±2.39** | 10.71±5.47 | 27.42±12.34 | 22.26±13.75 |
| *Random* | NLIN | Obs. MMD | **0.28±0.12** | 0.47±0.15 | 1.82±0.73 | 12.11±1.21 |
| | | Int. MMD | **1.45±0.07** | 1.88±0.22 | 3.52±1.03 | 14.15±2.34 |
| | | CF. MSE | **9.51±13.12** | 23.68±28.49 | 82.10±78.49 | 52.57±82.03 |
| | NADD | Obs. MMD | **0.19±0.05** | 0.31±0.14 | 2.09±0.60 | 12.63±1.10 |
| | | Int. MMD | **1.42±0.25** | 1.73±0.44 | 4.24±1.40 | 14.65±1.76 |
| | | CF. MSE | **20.13±57.52** | 44.76±86.02 | 124.82±275.09 | 54.29±83.68 |

Table 1: Mean±standard deviation of observational, interventional, and counterfactual queries of the ladder and random SCMs in nonlinear and nonadditive settings over 20 random initializations of the model and training data. The values are multiplied by 100 for clarity.

encoding, assuming the regression model fit the data well. We see this in the nonlinear setting, ANM performs well but struggles to outperform DCM, perhaps due to the complexity of fitting a neural network using classical models. Note thats since observational and interventional queries are inherently easier than counterfactual queries, it is natural that we observe smaller improvements over other baselines.

Our proposed DCM method exhibits superior performance compared to VACA and CAREFL, often by as much as an order of magnitude. The better performance of DCM over CAREFL may be attributed to the fact that DCM uses the causal graph, while CAREFL only relies on a non-unique causal ordering. In the case of VACA, the limited expressiveness of the GNN encoder-decoder might be the reason behind its inferior performance, especially when dealing with multivariable, multidimensional complex structural equations as considered here, a shortcoming that has also been acknowledged by the authors of VACA (Sánchez-Martin et al., 2022). Furthermore, VACA performs *approximate* inference, e.g. when performing a counterfactual query with $\mathrm{do}(X_1 = 2)$, the predicted counterfactual value for $X_1$ is not exactly 2 due to imperfections in the reconstruction. This design choice may result in downstream compounding errors, possibly explaining discrepancies in performance. To avoid penalizing this feature, all metrics are computed using downstream nodes from intervened nodes. To evaluate computational efficiency, we compared DCM with VACA and CAREFL using a single experimental setup (see Appendix D.1). The results suggest that DCM is significantly more computationally efficient than both VACA and CAREFL, with training taking 7 times less time. Lastly, the standard deviation of DCM is small relative to the other models, demonstrating relative consistency, which points to the robustness of our proposed approach.

## 5.2 Real Data Experiments I

We evaluate DCM on interventional real world data by evaluating our model on the electrical stimulation interventional fMRI data from (Thompson et al., 2020), using the experimental setup from (Khemakhem et al., 2021). The fMRI data comprises samples from 14 patients with medically refractory epilepsy, with time series of the Cingulate Gyrus (CG) and Heschl's Gyrus (HG). The assumed underlying causal structure is the bivarate graph CG → HG. Our interventional ground truth data comprises an intervened value of CG and an observed sample of HG. We defer the reader to (Thompson et al., 2020; Khemakhem et al., 2021) for a more thorough discussion of the dataset.

| Algorithm | Median Abs. Error | Mean Abs. Error |
|---|---|---|
| DCM | **0.5981 ± 9.5e-3** | **0.5779 ± 1.9e-3** |
| CAREFL | 0.5983 ± 3.0e-2 | 0.6004 ± 2.2e-2 |
| ANM | 0.6746 ± 4.8e-8 | 0.6498 ± 1.4e-6 |
| Linear SCM | 0.6045 ± 0 | 0.6042 ± 0 |

Table 2: Performances for interventional predictions on the fMRI dataset, of the form median/mean absolute error±standard deviation, using the mean over 10 random seeds. We do not include VACA due to implementation difficulties. The results for CAREFL, ANM, and Linear SCM are consistent with those observed by (Khemakhem et al., 2021). We note that the Linear SCM has zero standard deviation, as the ridge regression model does not vary with the random seed.

In Table 2, we note that the difference in performance is more minor than our synthetic results. We believe this is due to two reasons. Firstly, the data seems inherently close to linear, as exhibited by the relatively similar performance with the standard ridge regression model (Linear SCM). Secondly, we only have a single ground truth interventional value instead of multiple samples from the interventional distribution. As a result, we can only compute the absolute error based on this single value, rather than evaluating the maximum mean discrepancy (MMD) between the true and predicted interventional distributions. Specifically, in the above table, we compute the absolute error between the model prediction and the interventional sample for each of the 14 patients and report the mean/median. The availability of a single interventional introduces a possibly large amount of irreducible error, artificially inflating the error values. For more details on the error inflation, see Appendix C.5.

## 6 Concluding Remarks

We demonstrate that diffusion models, in particular, the DDIM formulation (which allows for unique encoding and decoding) provide a flexible and practical framework for approximating interventions (do-operator) and counterfactual (abduction-action-prediction) steps. Our approach, DCM, is applicable independent of the DAG structure. We find that empirically DCM outperforms competing methods in all three causal settings, observational, interventional, and counterfactual queries, across various classes of structural equations and graphs.

While not in scope of this paper, the proposed DCM approach can also be naturally extended to any setting where diffusion models are applicable like categorical data, images, etc. For higher dimensional spaces, we believe DCMs should scale nicely, as diffusion models are typically deployed for high-dimensional image settings and exhibit SOTA performance. Furthermore, we may leverage many of the optimization and implementation tricks, as diffusion models are a very active field of research.

The proposed method does come with certain limitations. For example, as with all the previously mentioned related approaches, DCM precludes unobserved confounding. The theoretical analyses require assumptions, not all of which are easy to test. However, our practical results suggest that DCM provides competitive empirical performance, even when some assumptions needed for our theoretical guarantees are violated.

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

## A  Missing Details from Section 4

**Notation.** For two sets $\mathcal{X}, \mathcal{Y}$ a map $f : \mathcal{X} \mapsto \mathcal{Y}$, and a set $S \subset \mathcal{X}$, we define $f(S) = \{f(x) : x \in S\}$. For $x \in \mathcal{X}$, we define $x + S = \{x + x' : x' \in \mathcal{X}\}$. For a random variable $X$, define $p_X(x)$ as the probability density function (PDF) at $x$. We use p.d. to denote positive definite matrices and $Jf|_x$ to denote the Jacobian of $f$ evaluated at $x$. For a function with two inputs $f(\cdot, \cdot)$, we define $f_x(Y) := f(x, Y)$ and $f_y(X) := f(X, y)$.

**Lemma 1.** *For $\mathcal{U}, \mathcal{Z} \subset \mathbb{R}$, consider a family of invertible functions $q_{x_{\mathrm{pa}}} : \mathcal{U} \to \mathcal{Z}$ for $x_{\mathrm{pa}} \in \mathcal{X}_{\mathrm{pa}} \subset \mathbb{R}^d$, then $\frac{dq_{x_{\mathrm{pa}}}}{du}(q_{x_{\mathrm{pa}}}^{-1}(z)) = c(z)$ for all $x_{\mathrm{pa}} \in \mathcal{X}_{\mathrm{pa}}$ if and only if $q_{x_{\mathrm{pa}}}$ can be expressed as*

$$q_{x_{\mathrm{pa}}}(u) = q(u + r(x_{\mathrm{pa}}))$$

*for some function $r$ and invertible $q$.*

*Proof.* First for the reverse direction, we may assume $q_{x_{\mathrm{pa}}}(u) = q(u + r(x_{\mathrm{pa}}))$. Then

$$\frac{dq_{x_{\mathrm{pa}}}}{du}(u) = \frac{dq}{du}(u + r(x_{\mathrm{pa}})).$$

Now plugging in $u = q_{x_{\mathrm{pa}}}^{-1}(z) = q^{-1}(z) - r(x_{\mathrm{pa}})$,

$$\frac{dq_{x_{\mathrm{pa}}}}{du}(q_{x_{\mathrm{pa}}}^{-1}(z)) = \frac{dq}{du}(q^{-1}(z) - r(x_{\mathrm{pa}}) + r(x_{\mathrm{pa}})) = \frac{dq}{du}(q^{-1}(z)) = c(z).$$

Therefore $\frac{dq_{x_{\mathrm{pa}}}}{du}(q_{x_{\mathrm{pa}}}^{-1}(z))$ is a function of $z$.

For the forward direction, assume $\frac{dq_{x_{\mathrm{pa}}}}{du}(q_{x_{\mathrm{pa}}}^{-1}(z)) = c(z)$. Define $s_{x_{\mathrm{pa}}} : \mathcal{Z} \to \mathcal{U}$ to be the inverse of $q_{x_{\mathrm{pa}}}$. By the inverse function theorem and by assumption,

$$\frac{ds_{x_{\mathrm{pa}}}}{dz}(z) = \frac{dq_{x_{\mathrm{pa}}}^{-1}}{dz}(z) = \frac{1}{\frac{dq_{x_{\mathrm{pa}}}}{du}(q_{x_{\mathrm{pa}}}^{-1}(z))} = \frac{1}{c(z)}$$

for all $x_{\mathrm{pa}}$. Since the derivatives of $s_{x_{\mathrm{pa}}}$ are equal for all $x_{\mathrm{pa}}$, by the mean value theorem, all $s_{x_{\mathrm{pa}}}$ are additive shifts of each other. Without loss of generality, we may consider an arbitrary fixed $x_{\mathrm{pa}_0} \in \mathcal{X}_{\mathrm{pa}}$ and reparametrize $s_{x_{\mathrm{pa}}}$ as

$$s_{x_{\mathrm{pa}}}(z) = s_{x_{\mathrm{pa}_0}}(z) - r(x_{\mathrm{pa}}).$$

Let $u = s_{x_{\mathrm{pa}}}(z)$. Then we have

$$s_{x_{\mathrm{pa}_0}}(z) = u + r(x_{\mathrm{pa}})$$
$$q_{x_{\mathrm{pa}}}(u) = z = q_{x_{\mathrm{pa}_0}}(u + r(x_{\mathrm{pa}})),$$

and we have the desired representation by choosing $q = q_{x_{\mathrm{pa}_0}}$. $\qquad\square$

**Theorem 1.** *Assume for $X \in \mathcal{X} \subset \mathbb{R}$ and exogenous noise $U \sim \mathrm{Unif}[0, 1]$, $X$ satisfies the structural equation: $X := f(X_{\mathrm{pa}}, U)$, where $X_{\mathrm{pa}} \in \mathcal{X}_{\mathrm{pa}} \subset \mathbb{R}^d$ are the parents of node $X$ and $U \perp\!\!\!\perp X_{\mathrm{pa}}$. Consider an encoder-decoder model with encoding function $g : \mathcal{X} \times \mathcal{X}_{\mathrm{pa}} \to \mathcal{Z}$ and decoding function $h : \mathcal{Z} \times \mathcal{X}_{\mathrm{pa}} \to \mathcal{X}$, $Z := g(X, X_{\mathrm{pa}})$, $\hat{X} := h(Z, X_{\mathrm{pa}})$. Assume the following conditions:*

1. *The encoding is independent of the parent values, $g(X, X_{\mathrm{pa}}) \perp\!\!\!\perp X_{\mathrm{pa}}$.*

2. *The structural equation $f$ is differentiable and strictly increasing with respect to $U$ for all $x_{\mathrm{pa}} \in \mathcal{X}_{\mathrm{pa}}$.*

3. *The encoding $g$ is invertible and differentiable with respect to $X$ for all $x_{\mathrm{pa}} \in \mathcal{X}_{\mathrm{pa}}$.*

*Then, $g(X, X_{\mathrm{pa}}) = \tilde{q}(U)$ for an invertible function $\tilde{q}$.*

*Proof.* First, we show that $g(X, X_{\mathrm{pa}}) = g(f(X_{\mathrm{pa}}, U), X_{\mathrm{pa}})$ is solely a function of $U$.

Since continuity and invertibility imply strict monotonicity, without loss of generality, assume $g$ is an strictly increasing function (if not, we may replace $g$ with $-g$ and use $h(-Z, X_{\mathrm{pa}})$). By properties of the composition of functions, $q_{x_{\mathrm{pa}}}(U) := g(f(x_{\mathrm{pa}}, U), x_{\mathrm{pa}})$ is also differentiable and strictly increasing with respect to $U$. Also, because of strict monotonicity it is also invertible.

By the assumption that the encoding $Z$ is independent of $X_{\mathrm{pa}}$,

$$Z = q_{x_{\mathrm{pa}}}(U) \perp\!\!\!\perp X_{\mathrm{pa}}. \tag{9}$$

Therefore the conditional distribution of $Z$ does not depend on $X_{\mathrm{pa}}$. Using the assumption that $U \perp\!\!\!\perp X_{\mathrm{pa}}$, for all $x_{\mathrm{pa}} \in \mathcal{X}_{\mathrm{pa}}$ and $z$ in the support of $Z$, by the change of density formula,

$$p_Z(z) = \frac{p_U(q_{x_{\mathrm{pa}}}^{-1}(z))}{\left|\frac{dq_{x_{\mathrm{pa}}}}{du}(q_{x_{\mathrm{pa}}}^{-1}(z))\right|} = \frac{\mathbb{1}\{q_{x_{\mathrm{pa}}}^{-1}(z) \in [0, 1]\}}{\frac{dq_{x_{\mathrm{pa}}}}{du}(q_{x_{\mathrm{pa}}}^{-1}(z))} = c_1(z). \tag{10}$$

The numerator follows from the fact that the noise is uniformly distributed. The term $\frac{dq_{x_{\mathrm{pa}}}}{du}(q_{x_{\mathrm{pa}}}^{-1}(z))$ is nonnegative since $q_{x_{\mathrm{pa}}}$ is increasing. Furthermore, since $p_Z(z) > 0$, the numerator in Eq. 10 is always equal to 1 and the

denominator must not depend on $X_{\mathrm{pa}}$,

$$\frac{dq_{x_{\mathrm{pa}}}}{du}(q_{x_{\mathrm{pa}}}^{-1}(z)) = c_2(z)$$

for some function $c_2$. From Lemma 1 (by replacing $a$ by $x_{\mathrm{pa}}$), we may express

$$q_{x_{\mathrm{pa}}}(u) = q(u + r(x_{\mathrm{pa}})) \tag{11}$$

for an invertible function $q$.

Next, since $Z \perp\!\!\!\perp X_{\mathrm{pa}}$, the support of $Z$ does not depend on $X_{\mathrm{pa}}$, equivalently the ranges of $q_{x_1}$ and $q_{x_2}$ are equal for all $x_1, x_2 \in \mathcal{X}_{\mathrm{pa}}$,

$$q_{x_1}([0,1]) = q_{x_2}([0,1]). \tag{12}$$

Applying Eq. 11 and the invertibility of $q$,

$$q([0,1] + r(x_1)) = q([0,1] + r(x_2))$$
$$[0,1] + r(x_1) = [0,1] + r(x_2)$$
$$[r(x_1), r(x_1) + 1] = [r(x_2), r(x_2) + 1]$$

Since this holds for all $x_1, x_2 \in \mathcal{X}_{\mathrm{pa}}$, we have $r(x_{\mathrm{pa}})$ is a constant function, or $r(x_{\mathrm{pa}}) \equiv r$. Thus let $\tilde{q}$ be $\tilde{q}(u) = q(u + r) = q_{x_{\mathrm{pa}}}(u)$, which is solely a function of $U$ for all $x_{\mathrm{pa}}$. For all $x_{\mathrm{pa}}$,

$$g(f(x_{\mathrm{pa}}, U), x_{\mathrm{pa}})) = q_{x_{\mathrm{pa}}}(U) = \tilde{q}(U) \implies g(f(X_{\mathrm{pa}}, U), X_{\mathrm{pa}}) = \tilde{q}(U), \tag{13}$$

for an invertible function $\tilde{q}$. This completes the proof. $\qquad\square$

**Corollary 1.** *Assume the conditions of Theorem 1. Furthermore, assume the encoder-decoder model pair $(g, h)$ satisfies: $h(g(X, X_{\mathrm{pa}}), X_{\mathrm{pa}}) = X$. Consider a factual sample pair $x^{\mathrm{F}} := (x, x_{\mathrm{pa}})$ where $x := f(x_{\mathrm{pa}}, u)$ and an intervention $\mathrm{do}(X_{\mathrm{pa}} := \gamma)$. Then, the counterfactual estimate, given by $h(g(x, x_{\mathrm{pa}}), \gamma)$ matches the true counterfactual outcome $x^{\mathrm{CF}} := f(\gamma, u)$.*

*Proof.* For the intervention $\mathrm{do}(X_{\mathrm{pa}} := \gamma)$, the true counterfactual outcome is $x^{\mathrm{CF}} := f(\gamma, u)$. By assumption, $h(g(x^{\mathrm{CF}}, x_{\mathrm{pa}}), x_{\mathrm{pa}}) = x^{\mathrm{CF}}$. Now since Eq. 13 holds true for all $X_{\mathrm{pa}}$ and $U$, it also holds for the factual and counterfactual samples. We have,

$$g(x, x_{\mathrm{pa}}) = g(f(x_{\mathrm{pa}}, u), x_{\mathrm{pa}}) = \tilde{q}(u) = g(f(\gamma, u), \gamma) = g(x^{\mathrm{CF}}, \gamma).$$

Therefore, the counterfactual estimate produced by the encoder-decoder model

$$h(g(x, x_{\mathrm{pa}}), \gamma) = h(g(x^{\mathrm{CF}}, \gamma), \gamma) = x^{\mathrm{CF}}.$$

This completes the proof. $\qquad\square$

**Corollary 2.** *Let $\gamma \geq 0$. Assume the conditions of Theorem 1. Furthermore, assume the encoder-decoder model pair $(g, h)$ under some metric $d$ (e.g., $\|\cdot\|_2$), has reconstruction error less than $\tau$: $d(h(g(X, X_{\mathrm{pa}}), X_{\mathrm{pa}}), X) \leq \tau$. Consider a factual sample pair $x^{\mathrm{F}} := (x, x_{\mathrm{pa}})$ where $x := f(x_{\mathrm{pa}}, u)$ and an intervention $\mathrm{do}(X_{\mathrm{pa}} := \gamma)$. Then, the error between the true counterfactual $x^{\mathrm{CF}} := f(\gamma, u)$ and counterfactual estimate given by $h(g(x, x_{\mathrm{pa}}), \gamma)$ is at most $\tau$. In other words, $d(h(g(x, x_{\mathrm{pa}}), \gamma), x^{\mathrm{CF}}) \leq \tau$.*

*Proof.* For the intervention $\mathrm{do}(X_{\mathrm{pa}} := \gamma)$, the counterfactual outcome is $x^{\mathrm{CF}} := f(\gamma, u)$. Since Eq. 13 holds true for all $X_{\mathrm{pa}}$ and $U$, it also holds for the factual and counterfactual samples. We have,

$$g(x, x_{\mathrm{pa}}) = g(f(x_{\mathrm{pa}}, u), x_{\mathrm{pa}}) = \tilde{q}(u) = g(f(\gamma, u), \gamma) = g(x^{\mathrm{CF}}, \gamma). \tag{14}$$

By the assumption on the reconstruction error of the encoder-decoder and 14,

$$d(h(g(x^{\mathrm{CF}}, \gamma), \gamma), x^{\mathrm{CF}}) \le \tau \tag{15}$$

$$d(h(g(x, x_{\mathrm{pa}}), \gamma), x^{\mathrm{CF}}) \le \tau. \tag{16}$$

$$\square$$

We now discuss a lemma that is an extension of Lemma 1 to higher dimensions.

**Lemma 2.** *For $\mathcal{U}, \mathcal{Z} \subset \mathbb{R}^m$, consider a family of invertible functions $q_{x_{\mathrm{pa}}} : \mathcal{U} \to \mathcal{Z}$ for $x_{\mathrm{pa}} \in \mathcal{X}_{\mathrm{pa}} \subset \mathbb{R}^d$, if $J q_{x_{\mathrm{pa}}}(q_{x_{\mathrm{pa}}}^{-1}(z)) = cA$ for all $x_{\mathrm{pa}} \in \mathcal{X}_{\mathrm{pa}}$ then $q_{x_{\mathrm{pa}}}$ can be expressed as*

$$q_{x_{\mathrm{pa}}}(u) = q(u + r(x_{\mathrm{pa}}))$$

*for some function $r$ and invertible $q$.*

*Proof.* Assume $J q_{x_{\mathrm{pa}}}(q_{x_{\mathrm{pa}}}^{-1}(z)) = cA$. By the inverse function theorem,

$$J q_{x_{\mathrm{pa}}}^{-1}|_z = \left( J q_{x_{\mathrm{pa}}}|_{q_{x_{\mathrm{pa}}}^{-1}(z)} \right)^{-1}. \tag{17}$$

Define $s_{x_{\mathrm{pa}}} : \mathcal{Z} \to \mathcal{U}$ to be the inverse of $q_{x_{\mathrm{pa}}}$. By assumption

$$\det J q_{x_{\mathrm{pa}}}|_{q_{x_{\mathrm{pa}}}^{-1}(z)} = \det J q_{x_{\mathrm{pa}}}^{-1}|_z = \det J s_{x_{\mathrm{pa}}}|_z = cA$$

for all $x_{\mathrm{pa}}$ and a constant $c$ and orthogonal matrix $A$. Since the Jacobian of $s_{x_{\mathrm{pa}}}$ is a scaled orthogonal matrix, $s_{x_{\mathrm{pa}}}$ is a conformal function. Therefore by Liouville's theorem, $s_{x_{\mathrm{pa}}}$ is a Möbius function (Blair, 2000), which implies that

$$s_{x_{\mathrm{pa}}}(z) = b_{x_{\mathrm{pa}}} + \alpha_{x_{\mathrm{pa}}} A_{x_{\mathrm{pa}}}(z - a_{x_{\mathrm{pa}}}) / \|z - a_{x_{\mathrm{pa}}}\|^{\varepsilon}, \tag{18}$$

where $A_{x_{\mathrm{pa}}}$ is an orthogonal matrix, $\varepsilon \in \{0, 2\}$, $a_{x_{\mathrm{pa}}} \in \mathbb{R}^m$, and $\alpha_{x_{\mathrm{pa}}} \in \mathbb{R}$. The Jacobian of $s_{x_{\mathrm{pa}}}$ is equal to $cA$ by assumption

$$J s_{x_{\mathrm{pa}}}|_z = \frac{\alpha_{x_{\mathrm{pa}}} A_{x_{\mathrm{pa}}}}{\|z - a_{x_{\mathrm{pa}}}\|^{\varepsilon}} \left( I - \varepsilon \frac{(z - a_{x_{\mathrm{pa}}})(z - a_{x_{\mathrm{pa}}})^T}{\|z - a_{x_{\mathrm{pa}}}\|^2} \right) = cA.$$

This imposes constraints on variables $\alpha$, $a$, and $\varepsilon$. Choose $z$ such that $z - a_{x_{\mathrm{pa}}} = kv$ for a unit vector $v$ and multiply by $A_{x_{\mathrm{pa}}}^{-1}$,

$$cAA_{x_{\mathrm{pa}}}^{-1} = \frac{\alpha_{x_{\mathrm{pa}}}}{\|kv\|^{\varepsilon}} \left( I - \varepsilon \frac{k^2 v v^T}{k^2 \|v\|^2} \right)$$

$$I = \varepsilon v v^T + \left( \frac{ck^{\varepsilon}}{\alpha_{x_{\mathrm{pa}}}} \right) AA_{x_{\mathrm{pa}}}^{-1}.$$

If $\varepsilon = 2$, choosing different values of $k$, implying different values of $z$, results in varying values of on the right hand side, which should be the constant identity matrix. Therefore we must have $\varepsilon = 0$. This also implies that $\alpha_{x_{\mathrm{pa}}} = c$ and $A = A_{x_{\mathrm{pa}}}$. This gives the further parametrization

$$s_{x_{\mathrm{pa}}}(z) = b_{x_{\mathrm{pa}}} - cA(z - a_{x_{\mathrm{pa}}}) = b'(x_{\mathrm{pa}}) + cAz$$

where $b'(x_{\mathrm{pa}}) = b_{x_{\mathrm{pa}}} - cAa_{x_{\mathrm{pa}}}$.

Without of loss of generality, we may consider an arbitrary fixed $x_{\mathrm{pa}_0} \in \mathcal{X}_{\mathrm{pa}}$,

$$s_{x_{\mathrm{pa}_0}}(z) - s_{x_{\mathrm{pa}}}(z) = r(x_{\mathrm{pa}}) := b'(x_{\mathrm{pa}_0}) - b'(x_{\mathrm{pa}}).$$

Let $u = s_{x_{\mathrm{pa}}}(z)$. Then we have

$$s_{x_{\mathrm{pa}_0}}(z) = u + r(x_{\mathrm{pa}})$$
$$q_{x_{\mathrm{pa}}}(u) = z = q_{x_{\mathrm{pa}_0}}(u + r(x_{\mathrm{pa}})),$$

and we have the desired representation by choosing $q = q_{x_{\mathrm{pa}_0}}$. $\square$

**Theorem 2.** *Assume for $X \in \mathcal{X} \subset \mathbb{R}^m$ and continuous exogenous noise $U \sim \mathrm{Unif}[0,1]^m$ for $m \geq 3$, and $X$ satisfies the structural equation*

$$X = f(X_{\mathrm{pa}}, U) \tag{6}$$

*where $X_{\mathrm{pa}} \in \mathcal{X}_{\mathrm{pa}} \subset \mathbb{R}^d$ are the parents of node $X$ and $U \perp\!\!\!\perp X_{\mathrm{pa}}$. Consider an encoder-decoder model with encoding function $g : \mathcal{X} \times \mathcal{X}_{\mathrm{pa}} \to \mathcal{Z}$ and decoding function $h : \mathcal{Z} \times \mathcal{X}_{\mathrm{pa}} \to \mathcal{X}$,*

$$Z := g(X, X_{\mathrm{pa}}), \quad \hat{X} := h(Z, X_{\mathrm{pa}}). \tag{7}$$

*Assume the following conditions:*

1. *The encoding is independent of the parent values, $g(X, X_{\mathrm{pa}}) \perp\!\!\!\perp X_{\mathrm{pa}}$.*

2. *The structural equation $f$ is invertible and differentiable with respect to $U$, and $Jf_{x_{\mathrm{pa}}}$ is p.d. for all $x_{\mathrm{pa}} \in \mathcal{X}_{\mathrm{pa}}$.*

3. *The encoding $g$ is invertible and differentiable with respect to $X$, and $Jg_{x_{\mathrm{pa}}}$ is p.d. for all $x_{\mathrm{pa}} \in \mathcal{X}_{\mathrm{pa}}$.*

4. *The encoding $q_{x_{\mathrm{pa}}}(U) := g(f(x_{\mathrm{pa}}, U), x_{\mathrm{pa}})$ satisfies $Jq_{x_{\mathrm{pa}}}|_{q_{x_{\mathrm{pa}}}^{-1}(z)} = c(x_{\mathrm{pa}})A$ for all $z \in \mathcal{Z}$ and $x_{\mathrm{pa}} \in \mathcal{X}_{\mathrm{pa}}$, where $c$ is a scalar function and $A$ is an orthogonal matrix.*

*Then, $g(f(X_{\mathrm{pa}}, U), X_{\mathrm{pa}}) = \tilde{q}(U)$ for an invertible function $\tilde{q}$.*

*Proof.* We may show that $g(X, X_{\mathrm{pa}}) = g(f(X_{\mathrm{pa}}, U), X_{\mathrm{pa}})$ is solely a function of $U$.

By properties of composition of functions, $q_{x_{\mathrm{pa}}}(U) := g(f(x_{\mathrm{pa}}, U), x_{\mathrm{pa}})$ is also invertible, differentiable. Since $Jf_{x_{\mathrm{pa}}}$ and $Jg_{x_{\mathrm{pa}}}$ are p.d. and $Jq_{x_{\mathrm{pa}}} = Jf_{x_{\mathrm{pa}}} Jg_{x_{\mathrm{pa}}}$, then $Jq_{x_{\mathrm{pa}}}$ is p.d. for all $x_{\mathrm{pa}} \in \mathcal{X}_{\mathrm{pa}}$ as well.

By the assumption that the encoding $Z$ is independent of $X_{\mathrm{pa}}$,

$$Z = q_{X_{\mathrm{pa}}}(U) \perp\!\!\!\perp X_{\mathrm{pa}}. \tag{19}$$

Therefore the conditional distribution of $Z$ does not depend on $X_{\mathrm{pa}}$. Using the assumption that $U \perp\!\!\!\perp X_{\mathrm{pa}}$, for all $x_{\mathrm{pa}} \in \mathcal{X}_{\mathrm{pa}}$ and $z$ in the support of $Z$, by the change of density formula,

$$p_Z(z) = \frac{p_U(q_{x_{\mathrm{pa}}}^{-1}(z))}{\left| \det Jq_{x_{\mathrm{pa}}}|_{q_{x_{\mathrm{pa}}}^{-1}(z)} \right|} = \frac{2^{-m} \mathbb{1}\{q_{x_{\mathrm{pa}}}^{-1}(z) \in [0,1]^m\}}{\det Jq_{x_{\mathrm{pa}}}|_{q_{x_{\mathrm{pa}}}^{-1}(z)}} = c_1(z). \tag{20}$$

The numerator follows from the fact that the noise is uniformly distributed. The determinant of the Jacobian term $Jq_{x_{\mathrm{pa}}}(q_{x_{\mathrm{pa}}}^{-1}(z))$ is nonnegative since $Jq_{x_{\mathrm{pa}}}$ is p.d. Furthermore, since $p_Z(z) > 0$, the numerator in Eq. 20 is always equal to $2^{-m}$ and the denominator must not depend on $X_{\mathrm{pa}}$,

$$\det Jq_{x_{\mathrm{pa}}}|_{q_{x_{\mathrm{pa}}}^{-1}(z)} = c_2(z) \tag{21}$$

for some function $c_2$. From our assumption, $Jq_{x_{\mathrm{pa}}}|_{q_{x_{\mathrm{pa}}}^{-1}(z)} = c(x_{\mathrm{pa}})A$ for an orthogonal matrix $A$ for all $z$. Applying this to Eq. 21,

$$\det Jq_{x_{\mathrm{pa}}}|_{q_{x_{\mathrm{pa}}}^{-1}(z)} = \det c(x_{\mathrm{pa}})A = c(x_{\mathrm{pa}}) = c_2(z),$$

which implies $c(x_{\mathrm{pa}}) \equiv c$ is a constant function, or $Jq_{x_{\mathrm{pa}}}|_{q_{x_{\mathrm{pa}}}^{-1}(z)} = cA$. By Lemma 2, we may express $q_{x_{\mathrm{pa}}}(u)$ as

$$q_{x_{\mathrm{pa}}}(u) = q(u + r(x_{\mathrm{pa}})) \tag{22}$$

for an invertible function $q$.

Next, since $Z \perp\!\!\!\perp X_{\mathrm{pa}}$, the support of $Z$ does not depend on $X_{\mathrm{pa}}$, equivalently the ranges of $q_{x_{\mathrm{pa}_1}}$ and $q_{x_{\mathrm{pa}_2}}$ are equal for all $x_1, x_2 \in \mathcal{X}_{\mathrm{pa}}$,

$$q_{x_1}([0,1]^m) = q_{x_2}([0,1]^m). \tag{23}$$

Applying Eq. 22 and the invertibility of $q$,

$$q([0,1]^m + r(x_1)) = q([0,1]^m + r(x_2))$$
$$[0,1]^m + r(x_1) = [0,1]^m + r(x_2)$$
$$[r(x_1), r(x_1) + 1]^m = [r(x_2), r(x_2) + 1]^m.$$

Since this holds for all $x_1, x_2 \in \mathcal{X}_{\mathrm{pa}}$, we have $r(x)$ is a constant, or $r(x) \equiv r$. Thus let $\tilde{q}$ be $\tilde{q}(u) = q(u+r) = q_{x_{\mathrm{pa}}}(u)$, which is solely a function of $U$ for all $x_{\mathrm{pa}}$. For all $x_{\mathrm{pa}}$,

$$g(f(x_{\mathrm{pa}}, U, x_{\mathrm{pa}})) = q_{x_{\mathrm{pa}}}(U) = \tilde{q}(U) \implies g(f(X_{\mathrm{pa}}, U), X_{\mathrm{pa}}) = \tilde{q}(U). \tag{24}$$

This completes the proof. $\square$

**Corollary 3.** *Assume the conditions of Theorem 2. Furthermore, assume the encoder-decoder model pair $(g, h)$ satisfies*

$$h(g(X, X_{\mathrm{pa}}), X_{\mathrm{pa}}) = X. \tag{25}$$

*Consider a factual sample pair $(x, x_{\mathrm{pa}})$ where $x := f(x_{\mathrm{pa}}, u)$ and an intervention $\mathrm{do}(X_{\mathrm{pa}} := \gamma)$. Then, the given by $h(g(x, x_{\mathrm{pa}}), \gamma)$ matches the true counterfactual outcome $x^{\mathrm{CF}} := f(\gamma, u)$.*

**Corollary 4.** *Let $\gamma \geq 0$. Assume the conditions of Theorem 2. Furthermore, assume the encoder-decoder model pair $(g, h)$ under some metric $d$ (e.g., $\|\cdot\|_2$), has reconstruction error less than $\tau$,*

$$d(h(g(X, X_{\mathrm{pa}}), X_{\mathrm{pa}}), X) \leq \tau. \tag{26}$$

*Consider a factual sample pair $(x, x_{\mathrm{pa}})$ where $x := f(x_{\mathrm{pa}}, u)$ and an intervention $\mathrm{do}(X_{\mathrm{pa}} := \gamma)$. Then, the error between the true counterfactual $x^{\mathrm{CF}} := f(\gamma, u)$ and counterfactual estimate $h(g(x, x_{\mathrm{pa}}), \gamma)$ is at most $\tau$, i.e., $d(h(g(x, x_{\mathrm{pa}}), \gamma), x^{\mathrm{CF}}) \leq \tau$.*

## B  Testing Independence between Parents and Encodings

We empirically evaluate the dependence between the encoding and parent values. We consider a bivariate nonlinear SCM $X_1 \to X_2$ where $X_2 = f(X_1, U_2) = X_1^2 + U_2$ and $X_1$ and $U_2$ are independently sampled from a standard normal distribution. We evaluate the HSIC between $X_1$ and the encoding of $X_2$. We fit our model on $n = 5000$ samples and evaluate the HSIC score on 1000 test samples from the same distribution. We compute a p-value using a kernel based independence test and compare our performance to ANM, a correctly specified model in this setting (Gretton et al., 2007). We perform this experiment 100 times. Given true independence, we expect the p-values to follow a uniform distribution. In the table below, we show some summary statistics of the p-values from the 100 trials, with the last row representing the expected values with true uniform p-values (which happens under the null hypothesis).

We provide p-values from the correctly specified ANM approach which are close to a uniform distribution, demonstrating that it is possible to have encodings that are close to independent. Although the p-values produced by our DCM approach are not completely uniform, the encodings do not to consistently reject the null hypothesis of independence.

|  | Mean | Std. Dev | 10% Quantile | 90% Quantile | Min | Max |
|---|---|---|---|---|---|---|
| DCM | 0.196 | 0.207 | 0.004 | 0.515 | 6e-6 | 0.947 |
| ANM | 0.419 | 0.255 | 0.092 | 0.774 | 3e-5 | 0.894 |
| True Uniform (null) | 0.500 | 0.288 | 0.100 | 0.900 | 1e-2 | 0.990 |

Table 3: Table of p-value descriptions.

These results demonstrate that it is empirically possible to obtain encodings independent of the parent variables. We further note that the ANM is correctly specified in this setting and DCM is relatively competitive despite being far more general.

## C  Missing Experimental Details

In this section, we provide missing details from Section 5.

### C.1  Model Hyperparameters

For all experiments in our evaluation, we hold the model hyperparameters constant. For DCM, we use $T = 100$ total time steps with a linear $\beta_t$ schedule interpolating between 1e-4 and 0.1, or $\beta_t = \left(0.1 - 10^{-4}\right) \frac{t-1}{T-1} + 10^{-4}$ for $t \in [T]$. To incorporate the parents' values and time step $t$, we simply concatenate the parent values and $t/T$ as input to the $\varepsilon_\theta$ model. We found that using the popular cosine schedule (Nichol & Dhariwal, 2021) resulted in worse empirical performance, as well as using a positional encoding for the time $t$. We believe the drop in performance from the positional encoding is due to the low dimensionality of the problem since the dimension of the positional encoder would dominate the dimension of the other inputs.

We also evaluated using classifier-free guidance (CFG) (Ho & Salimans, 2021) to improve the reliance on the parent values, however, we found this also decreased performance. We provide a plausible explanation that can be explained through Theorem 1. With Theorem 1, we would like our encoding $g(Y, X)$ to be independent of $X$, however using a CFG encoding $(1 + w)g(Y, X) - wg(Y, 0)$ would only serve to increase the dependence of $g(Y, X)$ on $X$, which is counterproductive to our objective.

For VACA, we use the default implementation[9], training for 500 epochs, with a learning rate of 0.005, and the encoder and decoder have hidden dimensions of size $[8, 8]$ and $[16]$ respectively, a latent vector dimension of 4, and a parent dropout rate of 0.2.

For CAREFL, we also use the default implementation[10] with the neural spline autoregressive flows (Durkan et al., 2019), training for 500 epochs with a learning rate of 0.005, four flows, and ten hidden units.

For ANM, we also use the default implementation to select a regression model. Given a set of fitted regression models, the ANM chooses the model with the lowest root mean squared error averaged over splits of the data. The ANM considers the following regressor models: linear, ridge, LASSO, elastic net, random forest, histogram gradient boosting, support vector, extra trees, k-NN, and AdaBoost.

### C.2  Details about the Additive Noise Model (ANM)

For a given node $X_i$ with parents $X_{\mathrm{pa}_i}$, consider fitting a regression model $\hat{f}_i$ where $\hat{f}_i(X_{\mathrm{pa}_i}) \approx X_i$. Using this regression model and the training dataset is sufficient for generating samples from the observational and interventional distribution, as well as computing counterfactuals.

**Observational/Interventional Samples.** Samples are constructed in topological order. For intervened nodes, the sampled value is always the intervened value. Non-intervened root nodes in the SCM are sampled from the empirical distribution of the training set. A new sample for $X_i$ is generated by sampling the parent value $X_{\mathrm{pa}_i}$ inductively and sampling $\hat{U}$ from the empirical residual distribution, and outputting $\hat{X}_i := \hat{f}_i(X_{\mathrm{pa}_i}) + \hat{U}$.

---

[9]https://github.com/psanch21/VACA
[10]https://github.com/piomonti/carefl/

**Counterfactual Estimation.** For a factual observation $x^{\mathrm{F}}$ and interventions on nodes $\mathcal{I}$ with values $\gamma$, the counterfactual estimate only differs from the factual estimate for all nodes that are intervened or downstream from an intervened node. We proceed in topological order. For each intervened node $i$, $\hat{x}_i^{\mathrm{CF}} := \gamma_i$. For each non-intervened node $i$ downstream from an intervened node, define $\hat{u}_i^{\mathrm{F}} := x_i^{\mathrm{F}} - \hat{f}_i(x_{\mathrm{pa}_i}^{\mathrm{F}})$, the residual and estimated noise for the factual sample. Let $\hat{x}_{\mathrm{pa}_i}^{\mathrm{CF}}$ be counterfactual estimates of the parents of $X_i$. Then $\hat{x}_i^{\mathrm{CF}} := \hat{f}_i(\hat{x}_{\mathrm{pa}_i}^{\mathrm{CF}}) + \hat{u}_i^{\mathrm{F}}$.

Therefore, for counterfactual queries, if the true functional equation $f_i$ is an additive noise model, then if $\hat{f}_i \approx f_i$, the regression model will have low counterfactual error. In fact, if $\hat{f}_i \equiv f_i$, then the regression model will have perfect counterfactual performance.

## C.3 Details about Random Graph Generation

The random graph is comprised of ten nodes. We randomly sample this graph by generating a random upper triangular adjacency matrix where each entry in the upper triangular half is each to $1$ with probability $30\%$. We then check that this graph is comprised of a single connected component (if not, we resample the graph). For a graphical representation, we provide an example in Figure 4.

## C.4 Query Evaluation Frameworks for Synthetic Data Experiments

**Observational Evaluation.** We generate $1000$ samples from both the fitted and true observational distribution and report the MMD between the two. Since DCM and the ANM use the empirical distribution for root nodes, we only take the MMD between nonroot nodes.

**Interventional Evaluation.** We consider interventions of individual nodes. For an intervention node $i$, we choose $20$ intervention values $\gamma_1, \ldots, \gamma_{20}$, linearly interpolating between the $10\%$ and $90\%$ quantiles of the marginal distribution of node $i$ to represent realistic interventions. Then for each intervention $\mathrm{do}(X_i := \gamma_j)$, we generate $100$ values from the fitted model and true causal model, $\hat{X}$ and $X^\star$ for the samples from the fitted model and true model respectively. Since the intervention only affects the descendants of node $i$, we subset $\hat{X}$ and $X^\star$ to include only the descendants of node $i$, and compute the MMD on $\hat{X}$ and $X^\star$ to obtain a distance $\delta_{i,j}$ between the interventional distribution for the specific node and interventional value. Lastly, we report the mean MMD over all $20$ intervention values and all intervened nodes. For the ladder graph, we choose to intervene on $X_2$ and $X_3$ as these are the farthest nodes upstream and capture the maximum difficulty of the intervention. For the random graph, we randomly select three non-sink nodes to intervene on. A formal description of our interventional evaluation framework is given in Algorithm 6.

---

**Algorithm 6** Evaluation of Interventional Queries

---
1: **for** each intervention node $i$ **do**
2:     $\gamma_1, \ldots, \gamma_{20}$ linearly interpolate $10\%$ to $90\%$ quantiles of node $i$
3:     **for** each intervention $\gamma_j$ generated above **do**
4:         Intervene $\mathrm{do}(X_i := \gamma_j)$
5:         Generate $100$ samples $\hat{X}$ from $\hat{M}$ and $X^\star$ from $M^\star$ of descendants of $i$
6:         $\delta_{i,j} \leftarrow \hat{\mathrm{MMD}}(\hat{X}, X^\star)$
7:     **end for**
8: **end for**
9: Output mean of all $\delta_{i,j}$

---

**Counterfactual Evaluation.** Similarly to interventional evaluation, we consider interventions of individual nodes and for node $i$, we choose $20$ intervention values $\gamma_1, \ldots, \gamma_{20}$, linearly interpolating between the $10\%$ and $90\%$ quantiles of the marginal distribution of node $i$ to represent realistic interventions. Then for each intervention $\mathrm{do}(X_i := \gamma_j)$, we generate $100$ nonintervened factual samples $x^{\mathrm{F}}$, and query for the estimated and true counterfactual values $\hat{x}^{\mathrm{CF}}$ and $x^{\mathrm{CF}}$ respectively. Similarly to before, $\hat{x}^{\mathrm{CF}}$ and $x^{\mathrm{CF}}$ only differ on the descendants of node $i$, therefore we only consider the subset of the descendants of node $i$. We compute the MSE $\delta_{i,j}$, since the counterfactual estimate and ground truth are point values, giving us an error for a specific node and interventional value. Lastly, we report the mean MSE over all $20$ intervention values and all intervened nodes. We use the same intervention nodes as in the interventional evaluation mentioned above. A formal description of our counterfactual evaluation framework is given in Algorithm 7 (Appendix C).

---

**Algorithm 7** Evaluation of Counterfactual Queries

---

1: **for** each intervention node $i$ **do**
2:     $\gamma_1, \ldots, \gamma_{20}$ linearly interpolate 10% to 90% quantiles of node $i$
3:     **for** each intervention $\gamma_j$ generated above **do**
4:         Generate 100 factual samples $x_1^{\mathrm{F}}, \ldots, x_{100}^{\mathrm{F}}$
5:         Intervene $\mathrm{do}(X_i := \gamma_j)$
6:         Using $\{x^{\mathrm{F}}\}_{k=1}^{100}$, compute counterfactual estimates $\{\hat{x}_k^{\mathrm{CF}}\}_{k=1}^{100}$ and true counterfactuals $\{x_k^{\mathrm{CF}}\}_{k=1}^{100}$ for all descendants of $i$
7:         $\delta_{i,j} \leftarrow \mathrm{MSE}(\{\hat{x}_k^{\mathrm{CF}}\}_{k=1}^{100}, \{x_k^{\mathrm{CF}}\}_{k=1}^{100})$
8:     **end for**
9: **end for**
10: Output mean of all $\delta_{i,j}$

---

### C.5 Explanation of Error Inflation in fMRI Experiments

In our fMRI experiments, we compute the absolute error on a single interventional sample and compute the absolute value. As an intuition for why we cannot hope to have errors close to zero and for why the errors are relatively much closer together, consider the following toy problem.

Assume $X_1, \ldots, X_n, Y_1, \ldots, Y_n \overset{\mathrm{iid}}{\sim} \mathcal{N}(\theta, 1)$ and we observe $X_1, \ldots, X_n$ as data. Consider the two following statistical problems:

1. Learn a distribution $\hat{D}$ such that samples from $\hat{D}$ and samples $Y_1, \ldots, Y_n$ achieves low MMD.

2. Learn an estimator that estimates $Y_1$ well under squared error.

In the first statistical problem, if $\hat{D}$ is a reasonable estimator, we should expect that more data leads to lower a MMD, for example the error may decay at a $1/n$ rate. We should expect MMD values close to zero, and the magnitude of the performance is directly interpretable.

In the second statistical problem, under squared error, the problem is equivalent to

$$\min_c \mathbb{E}_{Y_1 \sim \mathcal{N}(\theta,1)}(Y_1 - c)^2 = \min_c \theta^2 + 1 - 2c\theta + c^2.$$

The optimal estimator is the mean $\theta$ and achieves a squared error of 1. Of course in practice we do not know $\theta$, if we were to use the sample mean of $X_1, \ldots, X_n$, we would have an error of the order $1 + 1/n$. While we may still compare various estimators, e.g. sample mean, sample median, deep neural network, all the losses will be inflated by 1, causing the difference in performances to seem much more minute.

Our synthetic experiments hope to directly estimate the interventional distribution and computes the MMD between samples from the true and model's distributions, implying they are of the first statistical problem. The fMRI real data experiments aim to estimate a single intervention value from a distribution, implying they are of the second statistical problem. We should not expect these results to be very small, and the metric values should all be shifted by an intrinsic irreducible error.

## D  Additional Experiments

### D.1  Running Time Experiments

We present the training times in minutes for one seed on the ladder graph using the default implementation and parameters. For a fair comparison, these are all evaluated on a CPU. Note that ANM is the fastest as it uses standard regression models, and our proposed DCM approach is about 7-9x faster than CAREFL and VACA. The generation (inference) times for all the methods are in the order of 1 second. For VACA and CAREFL, we use the implementation provided by the respective authors.

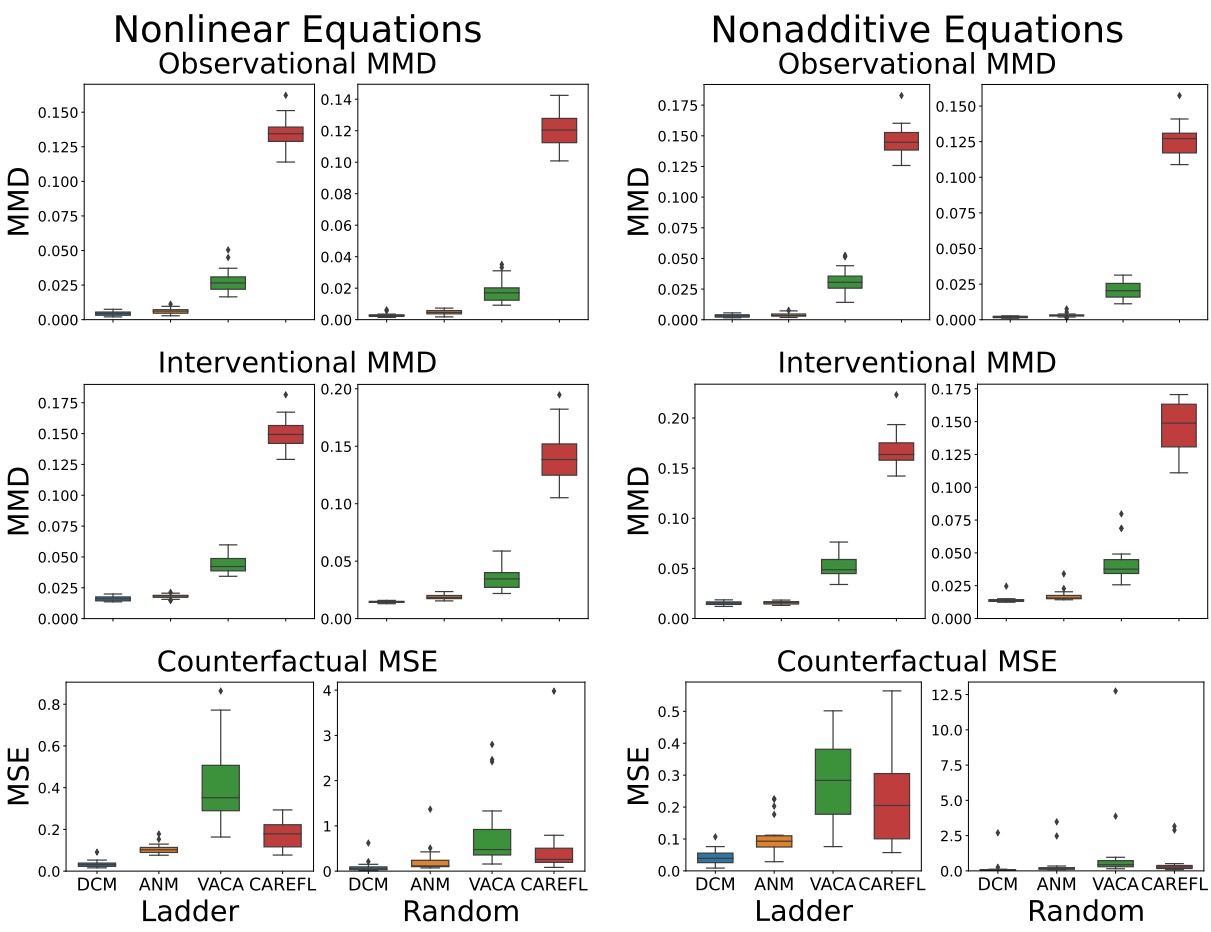

Figure 1: Left: Nonlinear setting (NLIN), Right: Nonadditive setting (NADD). Box plots of observational, interventional, and counterfactual queries of the ladder and random SCMs over 20 random initializations of the model and training data.

|  | DCM | ANM | VACA | CAREFL |
|---|---|---|---|---|
| Training Time (in minutes) | 15.3 | 4.3 | 142.8 | 110.5 |

Table 4: Table of training times for a single run.

## D.2 Additional Synthetic Experiments

In this section, we present additional experimental evidence showcasing the superior performance of DCM in addressing various types of causal queries. We consider four additional smaller graph structures, which we call the *chain*, *triangle*, *diamond*, and *Y* graphs (see Figure 9).

The exact equations are presented in Table 5. These functional equations were chosen to balance the signal-to-noise ratio of the covariates and noise to represent realistic settings. Furthermore, these structural equations were chosen *after* hyperparameter selection, meaning we did not tune DCM's parameters nor tune the structural equations after observing the performance of the models.

Consider a node with value $X$ with parents $X_{\mathrm{pa}}$ and exogenous noise $U$ where $X_{\mathrm{pa}} \perp\!\!\!\perp U$, and corresponding functional equation $f$ such that

$$X = f(X_{\mathrm{pa}}, U).$$

| SCM | | Nonlinear Case | Nonaddititive Case |
|---|---|---|---|
| *Chain* | $f_2(U_2, X_1)$ | $\exp(X_1/2) + U_2/4$ | $1/((U_2 + X_1)^2 + 0.5)$ |
| | $f_3(U_3, X_2)$ | $(X_2 - 5)^3/15 + U_3$ | $\sqrt{X_2 + |U_3|}/(0.1 + X_2)$ |
| *Triangle* | $f_2(U_2, X_1)$ | $2X_1^2 + U_2$ | $X_1/((U_2 + X_1)^2 + 1) + U_2/4$ |
| | $f_3(U_3, X_1, X_2)$ | $20/(1 + \exp(-X_2^2 + X_1)) + U_3$ | $(|U_3| + 0.3)(-X_1 + X_2/2 + |U_3|/5)^2$ |
| *Diamond* | $f_2(U_2, X_1)$ | $X_1^2 + U_2/2$ | $\sqrt{|X_1|}(|U_2| + 0.1)/2 + |X_1| + U_2/5$ |
| | $f_3(U_3, X_1, X_2)$ | $X_2^2 - 2/(1 + \exp(-X_1)) + U_3/2$ | $1/(1 + (|U_3| + 0.5)\exp(-X_2 + X_1))$ |
| | $f_4(U_4, X_2, X_3)$ | $X_3/(|X_2 + 2| + X_3 + 0.5) + U_4/10$ | $(X_3 + X_2 + U_4/4 - 7)^2 - 20$ |
| *Y* | $f_3(U_3, X_1, X_2)$ | $4/(1 + \exp(-X_1 - X_2)) - X_2^2 + U_3/2$ | $(X_1 - 2X_2 - 2)(|U_3| + 0.2)$ |
| | $f_4(U_4, X_3)$ | $20/(1 + \exp(X_3^2/2 - X_3)) + U_4$ | $(\cos(X_3) + U_4/2)^2$ |

Table 5: The equations defining the data generating process in the NLIN and NADD cases.

Further assuming the additive noise model, $X := f_1(X_{\mathrm{pa}}) + f_2(U)$. In this additive setting, since $X_{\mathrm{pa}} \perp\!\!\!\perp U$, we have

$$\mathrm{Var}\,[X] = \mathrm{Var}\,[f_1(X_{\mathrm{pa}}] + \mathrm{Var}\,[f_2(U)].$$

We choose $f_1$ and $f_2$ such that

$$0.05 \leq \frac{\mathrm{Var}\,[f_2(U)]}{\mathrm{Var}\,[f_1(X_{\mathrm{pa}}]} \leq 0.5,$$

representing the fact that the ratio of the effect of the noise to the parents is roughly approximate or smaller by an order of magnitude.

For the nonadditive case, we decompose the variance using the law of total variance,

$$\mathrm{Var}\,[X] = \mathbb{E}\,\mathrm{Var}\,[f(X_{\mathrm{pa}}, U) \mid U] + \mathrm{Var}\,\mathbb{E}[f(X_{\mathrm{pa}}, U) \mid U].$$

Similarly, we choose the functional equation $f$ such that $f$ satisfies

$$0.05 \leq \frac{\mathrm{Var}\,\mathbb{E}[f(X_{\mathrm{pa}}, U) \mid U]}{\mathbb{E}\,\mathrm{Var}\,[f(X_{\mathrm{pa}}, U) \mid U]} \leq 0.5$$

For all graphs, $U_i \overset{\mathrm{iid}}{\sim} \mathcal{N}(0, 1)$, and we choose $f$ such that $X_i = U_i$ if $X_i$ is a root node, i.e. $f$ is the identity function. Lastly, we normalize every node $X_i$ such that $\mathrm{Var}\,(X_i) \approx 1$. For the sake of clarity, we omit all normalizing terms in the formulas and omit functional equations for root nodes below.

**Results.** For these 4 graph structures (*chain*, *triangle*, *diamond*, and *Y*), in Table 6, we provide the performance of all evaluated models for observational, interventional, and counterfactual queries, averaged over 10 separate initializations

| SCM | | Metric | DCM ($\times 10^{-2}$) | ANM ($\times 10^{-2}$) | VACA ($\times 10^{-2}$) | CAREFL ($\times 10^{-2}$) |
|---|---|---|---|---|---|---|
| *Chain* | NLIN | Obs. MMD | 0.27±0.11 | **0.19±0.27** | 1.63±0.42 | 4.25±1.12 |
| | | Int. MMD | 1.71±0.27 | **1.70±0.47** | 10.10±1.81 | 8.63±0.52 |
| | | CF. MSE | **0.33±0.16** | 2.43±2.49 | 25.99±6.47 | 19.62±4.01 |
| | NADD | Obs. MMD | **0.22±0.16** | 1.51±0.43 | 1.53±0.41 | 3.48±1.06 |
| | | Int. MMD | **2.85±0.45** | 7.34±0.62 | 10.42±2.77 | 11.02±1.32 |
| | | CF. MSE | **75.84±1.65** | 88.56±2.63 | 98.82±4.16 | 105.80±11.04 |
| *Triangle* | NLIN | Obs. MMD | 0.16±0.11 | **0.12±0.07** | 3.12±0.93 | 4.64±1.03 |
| | | Int. MMD | **1.50±0.30** | 3.28±0.79 | 18.43±1.72 | 7.08±0.82 |
| | | CF. MSE | **1.12±0.26** | 9.80±1.70 | 178.69±16.45 | 41.85±19.58 |
| | NADD | Obs. MMD | **0.25±0.12** | 0.51±0.08 | 2.42±0.48 | 5.12±1.10 |
| | | Int. MMD | **2.81±0.21** | 5.54±0.60 | 11.09±0.85 | 4.17±0.44 |
| | | CF. MSE | **26.28±6.68** | 97.25±16.45 | 173.67±16.28 | 121.99±31.34 |
| *Y* | NLIN | Obs. MMD | **0.11±0.05** | 0.14±0.08 | 2.29±0.69 | 6.82±0.85 |
| | | Int. MMD | **1.23±0.08** | 1.40±0.14 | 9.50±0.96 | 14.97±1.29 |
| | | CF. MSE | **0.28±0.25** | 1.22±0.27 | 28.79±4.02 | 27.85±5.33 |
| | NADD | Obs. MMD | **0.21±0.15** | 1.00±0.19 | 1.51±0.44 | 3.39±0.58 |
| | | Int. MMD | **1.54±0.23** | 5.62±0.31 | 5.37±0.72 | 6.51±0.40 |
| | | CF. MSE | **33.45±2.06** | 47.55±2.56 | 60.67±3.24 | 52.41±6.69 |
| *Diamond* | NLIN | Obs. MMD | 0.22±0.10 | **0.13±0.07** | 2.77±0.45 | 8.28±1.29 |
| | | Int. MMD | 3.21±0.62 | **2.56±0.31** | 25.30±1.39 | 18.23±3.01 |
| | | CF. MSE | **14.74±6.09** | 32.02±37.74 | 138.65±12.33 | 607.62±241.21 |
| | NADD | Obs. MMD | **0.25±0.18** | 0.28±0.09 | 2.36±0.44 | 5.50±0.81 |
| | | Int. MMD | **1.88±0.23** | 4.40±0.52 | 12.54±0.89 | 16.34±1.72 |
| | | CF. MSE | **1.36±0.14** | 8.58±0.77 | 57.40±4.23 | 24.61±7.05 |

Table 6: Mean±standard deviation of observational, interventional, and counterfactual queries of four different SCMs in nonlinear and nonadditive settings over 10 random initializations of the model and training data. The values are multiplied by 100 for clarity.

of models and training data, with the lowest value in each row bolded. The values are multiplied by 100 for clarity. In Figure 2, we show the box plots for the same set of experiments.

The results here are similar to those observed with the larger graph structures in Table 1. DCM has the lowest error in 7 out of 12 of the nonlinear settings, with the correctly specified ANM having the lowest error in the remaining 5. Furthermore, DCM and ANM both typically have a lower standard deviation compared to the other competing methods. For the nonadditive settings, DCM demonstrates the lowest values for all 12 causal queries.

## D.3 Semi-Synthetic Experiments

To further evaluate the effectiveness of DCM, we explore a semi-synthetic experiment based on the Sachs dataset (Sachs et al., 2005). We use the real world graph from comprised of 11 nodes[11]. The graph represents an

---

[11]https://www.bnlearn.com/bnrepository/discrete-small.html#sachs

intricate network of signaling pathways within human T cells. The 11 nodes within this graph each correspond to one of the phosphorylated proteins or phospholipids that were examined in their study.

For our experiment, we sample data in a semi-synthetic manner. For the root nodes, we sample from the empirical marginal distribution. For non-root nodes, since the ground truth structural equations are unknown, we use a random neural network as the structural equation, as was done in Section 5. We report the performances in Table 7. We see that the performance improvements with our DCM approach corroborate our prior findings.

| SCM | | Metric | DCM $(\times 10^{-2})$ | ANM $(\times 10^{-2})$ | VACA $(\times 10^{-2})$ | CAREFL $(\times 10^{-2})$ |
|---|---|---|---|---|---|---|
| *Sachs* | NLIN | Obs. MMD | 0.31±0.15 | **0.21±0.04** | 0.53±0.24 | 7.30±0.95 |
| | | Int. MMD | **1.25±0.11** | 1.37±0.12 | 2.03±0.36 | 5.77±0.85 |
| | | CF. MSE | **0.72±0.19** | 4.72±1.71 | 17.71±9.23 | 9.59±2.52 |
| | NADD | Obs. MMD | **0.18±0.07** | 0.18±0.06 | 0.39±0.24 | 6.10±1.14 |
| | | Int. MMD | **1.42±0.26** | 1.86±0.51 | 2.21±0.98 | 5.29±2.07 |
| | | CF. MSE | **1.99±2.49** | 4.68±5.77 | 7.30±11.57 | 8.30±8.34 |

Table 7: Mean±standard deviation of observational, interventional, and counterfactual queries of the Sachs SCM in nonlinear (NLIN) and nonadditive (NADD) settings over 10 random initializations of the model and training data. The values are multiplied by 100 for clarity.

# E   Real Data Experiment II

In order to evaluate our DCM approach, we assess its performance in computing the Individual Treatment Effect (ITE), which is defined as $D_i := Y_i(1) - Y_i(0)$ where $Y_i(0) \in R$ is the potential outcome of unit $i$ when $i$ is assigned to the control group, and $Y_i(1)$ is the potential outcome when $i$ is assigned to the treatment group. Note that we do not observe $D_i$ for any unit $i$ as for each unit in the training data set, we observe either the outcome under control or the outcome under treatment, but never both.

Now, in non-simulated data, it is impossible to know the counterfactual ground-truth as this event did not happen. However, averaging the ITE over all individuals, i.e., $\mathbb{E}[Y(1) - Y(0)]$, provides us with an estimate of the average treatment effect (ATE), for which we have real-world datasets with known ground-truth. Therefore, to evaluate our approach, we first estimate the ITE for each sample by estimating each corresponding counterfactual outcome. We then average these to obtain the ATE, which we can then compare with the ground-truth ATE. Since our techniques are designed for constructing unit-level counterfactuals, and not ATE directly, the goal here is not to compare against other ATE estimation approaches, but rather demonstrate that the individual treatment effects computed through our counterfactuals are reasonably accurate.

The following experiments are based on popular datasets. Each experiment was repeated 10 times using the same model hyperparameters for each problem, without fine-tuning to the specific problem. For each dataset, we use the ground-truth graph structure from the literature and assign a DCM model to each non-root node. For root nodes, we can directly use the empirical distribution without the need to fit a particular model.

**Infant Health and Development Program Dataset.** The dataset aims at predicting the effect of specialized childcare on cognitive test scores of infants (Hill, 2011). The goal here is to see if a treatment (specialized childcare) improves the cognitive abilities of infants compared to infants that did not receive the treatment. The average treatment effect here is the difference between the expected cognitive test score under do(specialized child care = 1) and expected cognitive test score under do(specialized child care = 0). The ground-truth ATE (on cognitive test score) associated with this dataset is 4.021.

**Lalonde Dataset.** The dataset contains different demographic variables (e.g., gender, age, education etc.) with the goal to see if training programs increase earnings (LaLonde, 1986). The average treatment effect here is the difference

| Dataset | Algorithm | ATE (computed from ITE) | Relative Absolute Error (%) |
|---------|-----------|--------------------------|------------------------------|
| IHDP | DCM | 4.013 | 0.199% |
| IHDP | ANM | 3.957 | 1.592% |
| Lalonde | DCM | 1579.59 | 3.672% |
| Lalonde | ANM | 1516.20 | 7.538% |

Table 8: ATE estimation through computing unit-level counterfactuals. The ground-truth ATE provided with the IHDP and Lalonde datasets are 4.021 and 1639.80, respectively.

between the expected earnings under do(training program = 1) and expected earnings under do(training program = 0). The ground-truth ATE (on earnings) associated with this dataset is 1639.80.

**Results.** The results are summarized in Table 8. As noted above, the focus of this paper is on computing unit-level counterfactuals. The fact that the estimate of the ATE computed from via DCM approach matches the ground-truth ATE well can be seen as a sign that the unit-level counterfactuals were computed accurately. For baseline, we also show the results where we replace our DCM approach with additive noise model (ANM) approach for modeling the SCMs.

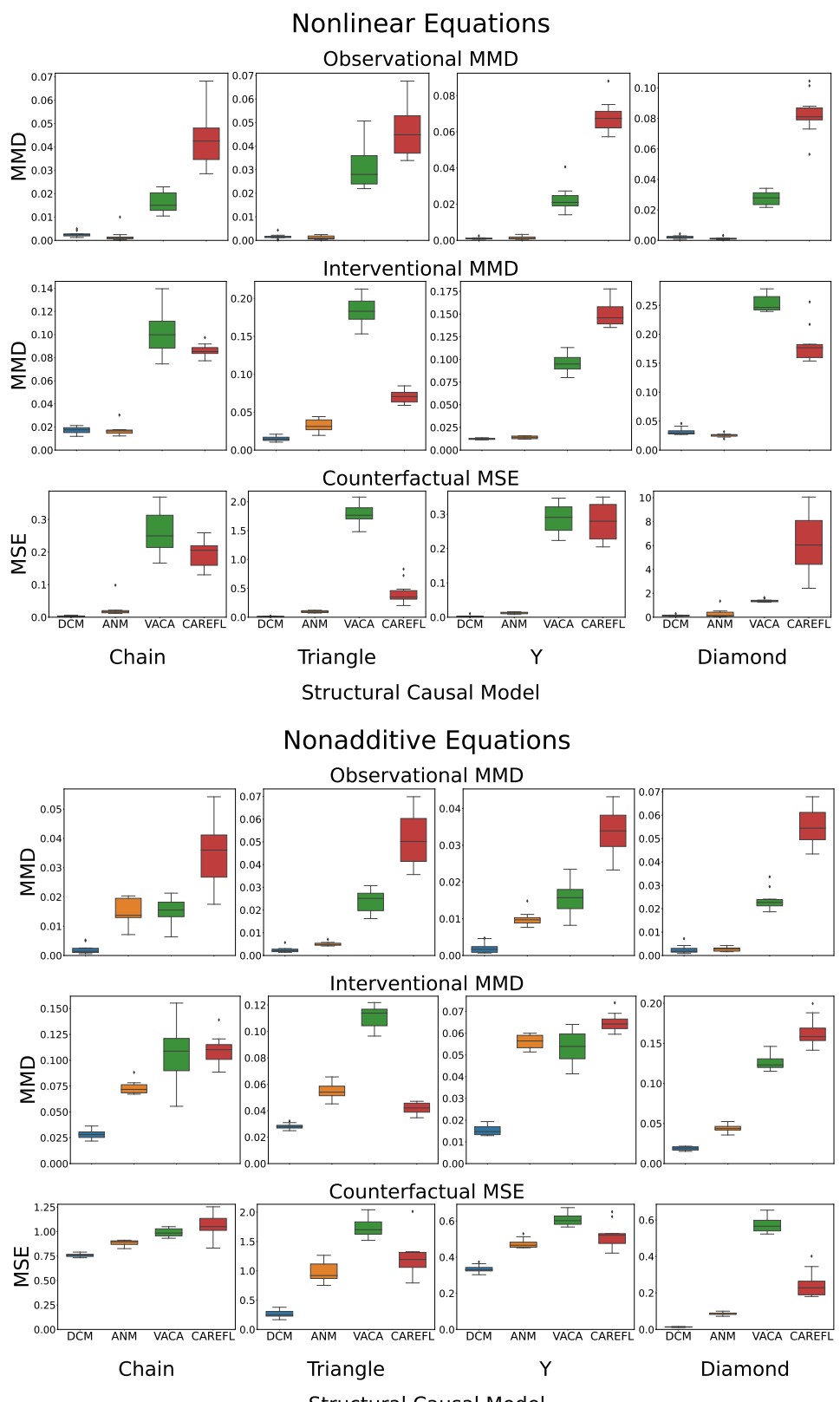

Figure 2: Top: Nonlinear setting (NLIN), Bottom: Nonadditive setting (NADD). Box plots of observational, interventional, and counterfactual queries of four different SCMs over 10 random initializations of the model and training data.

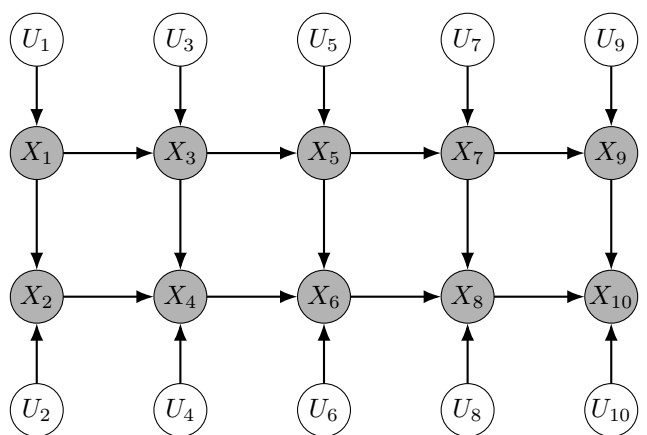

Figure 3: Ladder graph used in Section 5.

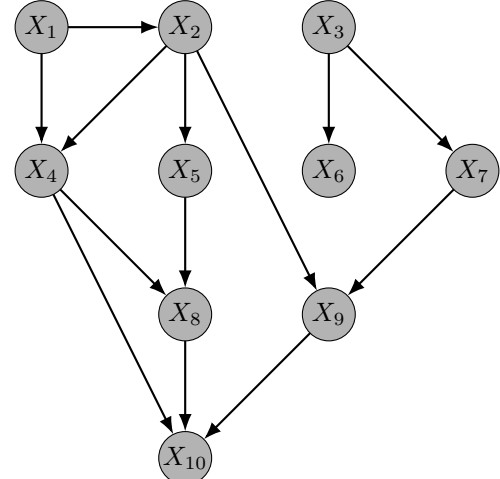

Figure 4: Example of a random graph used in Section 5, with exogenous noise nodes omitted for clarity.

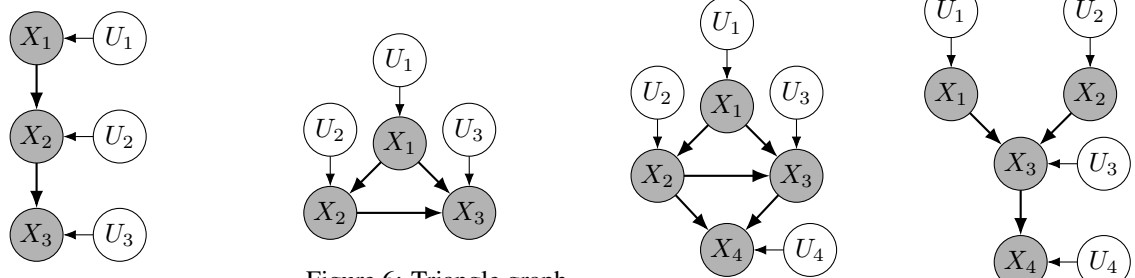

Figure 5: Chain graph.

Figure 6: Triangle graph.

Figure 7: Diamond graph.

Figure 8: Y graph.

Figure 9: Causal graphs used in our experiments in Appendix D.2.

