# OpenReview forum: "Modeling Causal Mechanisms with Diffusion Models for Interventional and Counterfactual Queries"
_TMLR — Accepted by TMLR_

### Review · Reviewer_4cS6 · 2024-08-04

**Summary Of Contributions:**

This paper provides the first attempt to use diffusion models in answering causal queries. The proposed method takes a causal graph as input, models each node in the causal graph as a diffusion model and learns how to sample from the data generating distribution by using diffusion models.

**Audience:**

Yes

**Broader Impact Concerns:**

no concerns

**Claims And Evidence:**

Yes

**Requested Changes:**

writing:
1. please formally introduce the problem of causal queries in the introduction. In the preliminaries, instead of explaining counterfactual queries in words, please provide a mathematical definition: do you mean the expected outcome here?
2. the first sentence of the second paragraph of the introduction is very confusing: "we focus on approximating SCM given observational data and the underlying causal DAG" This sentence sounds like you are learning the underlying DAG from data, but you are not. It is better to say along the lines of given a causal DAG, we focus on approximating the data generating distribution with diffusion models
3. the fourth to the last row of page 1 "which may be interpreted as a deterministic autoencoder model with latent variables": this is a causal paper, what do you mean by latent variables here? do you mean unmeasured confounding? how does this relate to the assumption of causal sufficiency that you subsequently made in the paper?
4. khemakhen et al. (2021) "their approach can be extended to more general DAGs": the proposed method scales exponentially in the number of variables in the DAG. This approach in general is not scalable and this is an overstatement.
5. in the discussion of assumption 2 "assumption 2 is always satisfied under additive noise model": is this true? f here is independent of U, so it seems that it would not satisfy the second half of the assumption where you assume that f is strictly increasing in U. "e.g., distinguishing between X:= X_pa + U and X:= X_pa - U": does this example contradict with the claim that assumption 2 is always satisfied under additive noise model? The two U's here can be thought of as having flipped distribution.

assumptions:
1. in the preliminaries end of the second paragraph "every SCM M entails a unique joint observational distribution satisfying the causal Markov assumption..." this seems to imply that the causal DAG is acyclic. Can you verify this and make it more explicit?
2. the Assumption 2 in Theorem 1 is too strong: I understand for diffusion models to work on this problem, you probably would require some form of assumption 2. However, I do not see how this assumption is easily satisfied in practice. Can the authors comment on the comparison to semi-parametric estimation methods which would not require this assumption? So long that I have a target parameter defined, I can estimate the data generating distribution nonparametrically (here we can also leverage the structure of the causal graph), perform a plug-in estimation, and subsequently perform a debiasing step to get asymptotically desired behavior.
3. In theorem 2, assumptions are listed without interpretation. What does it mean in practice for Jf_x_pa to be positive definite?

experiments:
1. ANM does not require the monotone assumption, can you verify whether your real-world data satisfies the monotone assumption listed in Assumption 2?
2. Can you provide some robustness check of your method when assumption 2 does not hold?
3. From the description of the real-world experiments, it is unclear to me what the format of the input data is and what the causal graph is. Is the graph simply CG->HG? The real-world experiment is not very convincing that this method can be useful in practice when compared with simpler benchmarks. A better case study is needed to illustrate the real-world use case of the proposed method.
4. Can the authors compare the proposed methods to statistical estimation methods? ANM is more commonly used as a discovery method than an estimation method. The other two benchmarks are not specialized in inference.

**Strengths And Weaknesses:**

strengths: the proposed method might have good potential in settings where sampling from the dataset is difficult, e.g., when the node of the causal graph is an image or text. The authors attempted to provide theoretical guarantees of the proposed methods under some relatively strong assumptions.

Weakness: The paper could also benefit from improved writing in the introduction to clarify the contribution and objective of the paper. the proposed method relies on strong assumptions about the underlying data generating distributions which might be unrealistic when the data type is complex (e.g., text/image). For simpler setups, e.g., the nodes of the causal graph are of numerical values, it is unclear whether the proposed method will outperform semi-parametric estimation methods. When the data generating process is complex, it is unclear how the causal DAG could be provided/constructed in the first place. A better example is needed. The numerical results on the real-world data seem a bit marginal.

Overall, I see the real-world potential use case of the proposed methods. However, this is hindered by the strong assumption made by the methods. A strong real-world case study will be needed for the paper to be publishable. I am happy to update the score if the authors can either 1) update their numerical results with better benchmarks and a stronger case study to support their contribution claim in the introduction, or 2) revise the scope of the contribution of the paper to acknowledge the limitation of the proposed methods at the beginning of the paper. The idea of using diffusion models to sample from DAG might be an idea that is of interest to a broader audience.

---

> ### Author Response · Authors · 2024-08-30
> **Response to Reviewer 4cS6 - Part 1**
>
> We thank the reviewer for the valuable feedback.
>
> For the reviewer’s main concern regarding "a strong real-world case study", we have included a new experimental section (Appendix E) that demonstrates how our scheme performs for the problem of estimating individual treatment effects. Also as we state (more explicitly now) in the Introduction, our primary objective is not to tackle a specific causal inference problem, but to propose a novel method for modeling Structural Causal Models (SCMs) using diffusion models, which provides a unified and flexible framework for answering a wide range of observational, interventional, and unit-level counterfactual queries.
>
> Please find below our responses to the other specific questions/concerns that you raise.
>
> > please formally introduce the problem of causal queries in the introduction. In the preliminaries, instead of explaining counterfactual queries in words, please provide a mathematical definition: do you mean the expected outcome here?
>
> We have now defined it in the second paragraph of the Introduction, stating that we are interested in counterfactual pertaining to a single unit of the population.
>
> > the first sentence of the second paragraph of the introduction is very confusing...
>
> Thanks. Fixed.
>
> > the fourth to the last row of page 1 "which may be interpreted as a deterministic autoencoder model with latent variables": this is a causal paper, what do you mean by latent variables here? do you mean unmeasured confounding? how does this relate to the assumption of causal sufficiency that you subsequently made in the paper?
>
> No, actually this has nothing to do with causal sufficiency assumption but rather was referring to the latent space which the encoders map to. We have reworded this sentence to avoid confusion.
>
> > khemakhen et al. (2021) "their approach can be extended to more general DAGs": the proposed method scales exponentially in the number of variables in the DAG. This approach in general is not scalable and this is an overstatement.
>
> We have removed this line as we do not think it is important.
>
> > in the discussion of assumption 2 "assumption 2 is always satisfied under additive noise model": is this true? f here is independent of U, so it seems that it would not satisfy the second half of the assumption where you assume that f is strictly increasing in U. "e.g., distinguishing between X:= X_pa + U and X:= X_pa - U": does this example contradict with the claim that assumption 2 is always satisfied under additive noise model? The two U's here can be thought of as having flipped distribution.
>
> We believe that there is some confusion due to overloading of $f$ in our text. We have fixed that now. In the additive noise model, $f(X_\text{pa},U) := f'(X_\text{pa}) + U$ for some $f'$. So $f$ is not independent of $U$ ($f'$ is). So as you see in the additive noise model, $f(X,X_\text{pa})$ is trivially both differentiable and strictly increasing with respect to $U$.
>
> > in the preliminaries end of the second paragraph "every SCM M entails a unique joint observational distribution satisfying the causal Markov assumption..." this seems to imply that the causal DAG is acyclic. Can you verify this and make it more explicit?
>
> Yes, we emphasize that at the start of this paragraph.
>
> > the Assumption 2 in Theorem 1 is too strong: I understand for diffusion models to work on this problem, you probably would require some form of assumption 2. However, I do not see how this assumption is easily satisfied in practice.
>
> Assumption 2 is not on the diffusion model, but the underlying generation mechanism $f$. As we explain in the Remark (2) following Theorem 1, Assumption 2 is both standard in similar results in the literature, and somewhat necessary for any identifiability result regardless of whether we use diffusion model or any other technique to model SCMs. Also, as we note later, our scheme works even when this assumption is violated.
>
> > Can the authors comment on the comparison to semi-parametric estimation methods which would not require this assumption? So long that I have a target parameter defined, I can estimate the data generating distribution nonparametrically (here we can also leverage the structure of the causal graph), perform a plug-in estimation, and subsequently perform a debiasing step to get asymptotically desired behavior.
>
> This is an interesting question. Our goal is to construct a setup where once you train your diffusion models, you can answer any number of (unit-level) counterfactual queries without any need for retraining. This is different from setups where you have a single target parameter defined. Additionally, even in the potential outcomes framework, individual treatment effect (ITE), is not identifiable without strong additional assumptions. For example, one can construct data-generating processes with the same distribution of the observed data, but a different ITE (see e.g., Example 1 Kunzel et al. 2019: https://arxiv.org/pdf/1706.03461)

---

> > ### Author Response · Authors · 2024-08-30
> > **Response to Reviewer 4cS6 - Part 2**
> >
> > > In theorem 2, assumptions are listed without interpretation.
> >
> > The interpretation of Assumptions 1,2, and 3 in Theorem 2 are identical to that of corresponding assumptions in Theorem 1. Assumption 4 in Theorem 2 though a technical assumption that does not have a clear interpretation. This assumption, as we explain in the paragraph before Section 5 though helps us avoid the negative result of Nasr-Esfahany & Kiciman (2023).
> >
> > > What does it mean in practice for Jf_x_pa to be positive definite?
> >
> > Since $f(X_{\text{pa}},U)$ is a vector-valued function, the derivative of $f$ with respect to $U$ is the Jacobian functional $J f(X_{\text{pa}})$, and $J f_{x_\text{pa}} \in \mathbb{R}^{m \times m}$ is the matrix obtained by setting $X_{\text{pa}} = x_{\text{pa}}$ in $J f(X_{\text{pa}})$. The positive definiteness (implying positive determinant) is needed for change of density trick in the proof. Another way of restating this positive-definiteness of $J f_{x_\text{pa}}$ is by requiring $f(X_{\text{pa}},U)$ to be strictly monotone in $U$ for all $x_{\text{pa}} \in \mathcal{X}_{\text{pa}}$.
> >
> > Note that this is trivially true for additive noise models, $f(X_{\text{pa}},U) := f'(X_{\text{pa}}) + U$.
> >
> > > ANM does not require the monotone assumption, can you verify whether your real-world data satisfies the monotone assumption listed in Assumption 2?
> >
> > As we explained in the answer above, ANM's are monotonic in $U$ (Assumption 2).
> >
> > > Can you provide some robustness check of your method when assumption 2 does not hold?
> >
> > Actually, we already do that in our synthetic data experiments. For example, if you look at Appendix D.3 (Table 6), you will notice that the ground-truth mechanisms described there are not differentiable or monotone in noise variables (violating Assumption 2). However, as our results reflect in Table 7, our DCM approach manages to perform well on all observational, interventional, and counterfactual queries.
> >
> > > From the description of the real-world experiments, it is unclear to me what the format of the input data is and what the causal graph is. Is the graph simply CG->HG?
> >
> > Yes, in Section 5.2 that is the graph.
> >
> > > The real-world experiment is not very convincing that this method can be useful in practice when compared with simpler benchmarks. A better case study is needed to illustrate the real-world use case of the proposed method.
> >
> > 1. We do have a semi-synthetic experiment already in Appendix D.2 based on Sachs dataset.
> > 2. We have also added an entire new set of experiments in Appendix E on a real-world use case of estimating unit-level counterfactuals (which leads to estimating individual treatment effect).
> >
> > > Can the authors compare the proposed methods to statistical estimation methods? ANM is more commonly used as a discovery method than an estimation method. The other two benchmarks are not specialized in inference.
> >
> > The main goal of the paper is to provide a new and flexible way of modeling SCMs using diffusion models. Once these diffusion models are trained, we show that it can be used to answer any number of observational, interventional, and (unit-level) counterfactual queries, which form the three layers of causal queries with increasing complexity in Pearl's causal hierarchy (Pearl, 2009a).
> >
> > Prior to our paper, VACA and CAREFL were SOTA techniques with the same goal. ANMs are simple models that work well in practice. Therefore, we focused on these as baselines.
> >
> > Statistical estimation theory is generally focused on specific problems like ATE/CATE estimation.  Even with counterfactuals, we are interested in unit-level counterfactuals, for which SCM paradigm provides a three-step procedure: Abduction, Action, and Prediction.
> >
> > Given our focus on developing an approach that works well on a broad class of important causal inference problems, we believe that we have compared against the right set of baselines.

---

> > > ### Comment · Reviewer_4cS6 · 2024-09-05
> > > **a follow-up question**
> > >
> > > I find the ITE model more convincing.
> > >
> > > Here is a point that needs to be clarified further:
> > > In Theorem 1, I agree that f(X,U) = f'(X) + U is monotonically increasing in U. My confusion is on the second half of the explanation:
> > >
> > > You assumed that the noise U is uniform~[0,1]: I think this is how you claim that you can distinguish between X=X_pa-U and  X=X_pa + U. As in your D.3 experiments when the noise is generated from N(0,1), I do not think you can claim this. This sentence is still very confusing.
> > >
> > > In your updated body, you wrote: "Assumption 2 is always satisfied under the additive noise model (i.e., f(Xpa,U) := f′(Xpa) + U) and post nonlinear models." How is this satisfied in the post-nonlinear models?
> > > Note that in your response you wrote ``you will notice that the ground-truth mechanisms described there are not differentiable or monotone in noise variables''

---

> > > > ### Author Response · Authors · 2024-09-06
> > > > **Response to Follow-up Question**
> > > >
> > > > We are glad that to have addressed the reviewers' main concern with the new set of experiments. We now address the follow-up questions.
> > > >
> > > > >You assumed that the noise U is uniform~[0,1]: I think this is how you claim that you can distinguish between X=X_pa-U and X=X_pa + U. As in your D.3 experiments when the noise is generated from N(0,1), I do not think you can claim this. This sentence is still very confusing.
> > > >
> > > > We make the strictly increasing Assumption 2 in Theorem 1 to distinguish cases like $f(X_{\text{pa}},U):=X_{\text{pa}} + U$ vs. $f(X_{\text{pa}},U):=X_{\text{pa}} - U$, which otherwise will be indistinguishable for symmetric distributions $U$. Since, for any fixed value of $X_{\text{pa}}$, $X_{\text{pa}} - U$ is not increasing in $U$, it is automatically eliminated by Assumption 2 in Theorem 1. The fact that $U$ is uniform in $[0,1]$ is not critical for this part. We modified the text in Remark (2) to clarify this.
> > > >
> > > > Also, as a side remark, it is possible to modify the theoretical arguments to work under the assumption that $f$ is *strictly decreasing* in $U$, in which case $f(X_{\text{pa}},U):=X_{\text{pa}} + U$ is eliminated from consideration.
> > > >
> > > > Now while the assumptions in the theorems are needed for theoretical proof to work, what we show in experiments is that even when the assumptions are violated our DCM scheme works. For example, in Table 6, Chain graph, nonadditive case, we define, the generating function for the node $X_3$ as $\sqrt{X_2 + |U_3|}/(0.1 + X_2)$ which is neither differentiable nor strictly increasing in $U_3$.
> > > >
> > > > Finally, the noise distribution in our experiments is $N(0,1)$, however, that in itself plays does not an important role, and we get very similar performance improvements with uniform noise in the synthetic experiments.
> > > >
> > > > > In your updated body, you wrote: "Assumption 2 is always satisfied under the additive noise model (i.e., f(Xpa,U) := f′(Xpa) + U) and post nonlinear models." How is this satisfied in the post-nonlinear models? Note that in your response you wrote ``you will notice that the ground-truth mechanisms described there are not differentiable or monotone in noise variables'
> > > >
> > > > Thanks, we realize that the line about post-nonlinear models requires further clarification.
> > > > Zhang & Hyvarinen 2012 (see their Eqn. 2) defined post-nonlinear models as $f(X_{\text{pa}},U):=f_2(f_1(X_{\text{pa}})+U)$ where $X_\text{pa}$ and $U$ are independent, function $f_1$ is nonconstant, and $f_2$ is invertible. The invertibility assumption on $f_2$ implies that $f$ is strictly increasing (or decreasing) in $U$. The differentiability of $f_2$ is not explicitly stated in the definition, but is needed in their results of identifiability, see their Assumption A1, which gives the differentiability of $f$ in $U$.
> > > > We have modified the Remark 2 in our paper to make this clearer.
> > > >
> > > > The experimental results (like in Section D.3) show the robustness of our DCM approach to the theoretical assumptions, i.e., our schemes work practically even when the assumptions are violated.

---

> > > > > ### Comment · Reviewer_4cS6 · 2024-09-10
> > > > > **response**
> > > > >
> > > > > I think the reason why I was confused about the language "distinguishing the cases X=X_pa-U and X=X_pa +U" is because, under symmetric U, these two distributions are not distinguishable. You might want to change the phrasing to something like monotonic assumption forces the causal structure to take the additive noise form X=X_pa + U. However, this is so standard in ANM that you might want to just eliminate this sentence.
> > > > >
> > > > > Updated my score.

---

> > > > > > ### Author Response · Authors · 2024-09-10
> > > > > > **Response to Follow-up Question**
> > > > > >
> > > > > > Thanks for the suggestion. We will remove the sentence in the revised version.

---

### Review · Reviewer_A998 · 2024-08-08

**Summary Of Contributions:**

The paper proposes diffusion-based causal models (DCMs), which use diffusion models to approximate the functions in a structural causal model, given a DAG and observational data. The intuition is similar to the classic approach of using regression methods to approximate these functions, but instead using more expressive and flexible diffusion models based on neural networks. DCMs allow answering interventional and counterfactual queries by constructing a Markov chain of samples from post-intervention distributions. The paper also offers bounds on the error of counterfactual estimates in more general encoder-decoder approaches.

**Audience:**

Yes

**Claims And Evidence:**

Yes

**Requested Changes:**

(2. is critical, the others are suggested)
1. Go carefully through the intro again---it seemed to have more mistakes and less clear/polished writing than other parts of the paper (though it was more clear to me the second time through, after understanding more about diffusion models). For example:
    - "... between a Gaussian and a density." seems to be missing some words
    - "... these are first error bounds..." seems to be missing "the"
    - "In contrast to our work, while Javaloy et al. (2023) focuses on modeling the whole causal graph as one model." Is the word "while" unnecessary, or is some other part of the sentence missing?
2. "Note that causal sufficiency is the minimal necessary set of assumptions for causal reasoning from observational data alone." Isn't this just one assumption and not a set? More importantly, this claim needs more explanation, context, or justification. Why doesn't something like latent causal discovery count as causal reasoning?
3. "For example, our DCM approach trains approximately seven times faster than CAREFL and nine times faster than VACA." This reads like a very general claim, but it's actually only supported by a seed on a single ground truth. Weaken the claim or add more support.

**Strengths And Weaknesses:**

**Strengths**:
- good clarity of writing, especially after Section 1
- ample theoretical support along with concise proof-of-concept experiments
- fairly intuitive approach but with a more expressive class of models than previous papers

**Weaknesses**:
- Section 1 is less consistently well-written than the rest of the paper

---

> ### Author Response · Authors · 2024-08-30
> **Response to Reviewer A998**
>
> We thank the reviewer for the valuable feedback. Please find below our responses to the specific concerns that you mention.
>
> > Go carefully through the intro again---it seemed to have more mistakes and less clear/polished writing than other parts of the paper (though it was more clear to me the second time through, after understanding more about diffusion models)
>
> Thanks for the suggestion. We have made the suggested changes along with few other edits to improve readability.
>
> > "Note that causal sufficiency is the minimal necessary set of assumptions for causal reasoning from observational data alone." Isn't this just one assumption and not a set? More importantly, this claim needs more explanation, context, or justification. Why doesn't something like latent causal discovery count as causal reasoning?
>
> We have edited this line to make this clearer. Causal sufficiency is necessary even for causal discovery in the plain observational data setting. With just observational data, only the construction of a Markov equivalence class is guaranteed. Access to instrumental variables or experimental data, which we do not assume, may help relax these assumptions.
>
> > "For example, our DCM approach trains approximately seven times faster than CAREFL and nine times faster than VACA." This reads like a very general claim, but it's actually only supported by a seed on a single ground truth. Weaken the claim or add more support.
>
> Thanks. We have restated the statement to clarify this finding is based only on a single experiment.

---

### Review · Reviewer_Sktb · 2024-08-16

**Summary Of Contributions:**

This paper proposes a method for learning and inference of structural causal models based on the diffusion models.
The proposed algorithm for training works given the causal graph and observational data.
Their trained models are able to answer causal queries including observational, interventional,
and counterfactual queries.
Empirical evaluations demonstrate that their proposed methods superior to state-of-the-art methods for causal queries.
Their contributions also include a framework for theoretical analysis of encoder-decoder-based structural causal models.

**Audience:**

Yes

**Claims And Evidence:**

Yes

**Requested Changes:**

If the authors can address any of the points listed as weakness above,
I would appreciate it.
However,
I am not strongly opposed to accepting this paper as it is.

**Strengths And Weaknesses:**

Strength:
* The positioning of the study and contributions are clearly explained along with many references.
* The effectiveness of the proposed method has been verified by both artificial and real data experiments, and is sufficiently convincing.
* The proposed method is natural and generic and should be of interest to the machine learning community.

Weakness:
* It is not clear whether the theoretical analysis in Section 4 is applicable to the proposed diffusion-based approach in Section 3.
For example, there seems to be no discussion on whether assumptions 2 and 3 of Theorem 1 are valid in the diffusion model.
Assumption 1 has also only been experimentally verified, and there is no theoretical evidence that the diffusion model satisfies this assumption.
* I could not discern the logical flow that led to the proposal in this paper.
In particular,
in dealing with structural causal models,
I would like to know why the diffusion model was chosen among various encoder-decoder models that are applicable,
and what challenges it addresses that other models cannot.
Since the theoretical analysis part of Section 4 can be applied to arbitrary encoder-decoder models,
I do not believe this supports the usefulness of the diffusion model.
The fact that each node can be trained in parallel is also not limited to diffusion models.

---

> ### Author Response · Authors · 2024-08-30
> **Response to Reviewer Sktb**
>
> Thank you very much for your encouraging comments. Please find below our responses to the specific weaknesses that you mention.
>
> > It is not clear whether the theoretical analysis in Section 4 is applicable to the proposed diffusion-based approach in Section 3. For example, there seems to be no discussion on whether assumptions 2 and 3 of Theorem 1 are valid in the diffusion model. Assumption 1 has also only been experimentally verified, and there is no theoretical evidence that the diffusion model satisfies this assumption.
>
> Theorem 1, Assumption 2 is only on the ground-truth structural function $f$. No assumption on the diffusion model is made there. In fact, as we explain in the Remark (2) following Theorem 1, Assumption 2 is both standard in similar results in the literature, and somewhat necessary for any identifiability result regardless of whether we use diffusion model or any other technique to model SCMs.
>
> Theorem 1, Assumption 3 is on the diffusion model. For the diffusion model associated with a particular node $i$, the encoding is defined in Algorithm 2, and the invertibility and differentiability boils down to the design of the network $\varepsilon_\theta^i$. The same holds for Assumption 1. A general theoretical proof of independence is hard, hence, we resorted to experimental verification (Appendix B). Also, as we note in Footnote 3, one could modify the diffusion model training objective to help achieve this independence.
>
>
> > I could not discern the logical flow that led to the proposal in this paper. In particular, in dealing with structural causal models, I would like to know why the diffusion model was chosen among various encoder-decoder models that are applicable, and what challenges it addresses that other models cannot. Since the theoretical analysis part of Section 4 can be applied to arbitrary encoder-decoder models, I do not believe this supports the usefulness of the diffusion model. The fact that each node can be trained in parallel is also not limited to diffusion models.
>
> There are two reasons to prefer diffusion models: 1) Practically, they are known to be highly performant for generative tasks, 2) Theoretically, they ensure that the encoding $g(X,X_\text{pa})$ is the same dimension as the input $X$. This is essential for the invertibility (in $X$) Assumption 3 in Theorems 1 and 2. Many encoder-decoder techniques (e.g., typical autoencoders) encode to a lower dimension latent space, which will violate this invertibility assumption.

---

### Author Response · Authors · 2024-08-30
**Common Response**

We thank all the reviewers for their comments and suggestions. We have directly addressed all their concerns. We also have updated the paper based on their suggestions. In the new version, the text marked in red reflect all the updates.

---

### Decision · Action_Editor_KQKc · 2024-10-15

**Recommendation:** Accept with minor revision

**Comment:**

The reviews were detailed and there was significant discussion between the reviewers and the authors. There are no further comments that came up in the discussion and all reviewers are supportive of the paper being accepted by TMLR. The authors should update the manuscript to remove the highlighting on the amendments and the paper should be ready for publication.

**Audience:**

The paper will be of interest to a sub-community within the broader ML research community, i.e. those with interest causal inference.

**Claims And Evidence:**

The claims made in the paper are supported. The paper has theory with proofs. There is also some empirical evidence, this is primarily proof of concept, but that is expected for a paper that establishes theoretical results.